# Polygenic contribution to the relationship of loneliness and social isolation with schizophrenia

Álvaro Andreu-Bernabeu[1,2,15], Covadonga M. Díaz-Caneja [1,2,3,4], Javier Costas[5], Lucía De Hoyos [1,2], Carol Stella[1,2], Xaquín Gurriarán[1,2,5], Clara Alloza[1,2], Lourdes Fañanás [3,6], Julio Bobes [3,7], Ana González-Pinto[3,8], Benedicto Crespo-Facorro[3,9], Lourdes Martorell[3,10], Elisabet Vilella [3,10], Gerard Muntané[3,10], Juan Nacher[3,11], María Dolores Molto[3,12,13], Eduardo Jesús Aguilar [3,13,14], Mara Parellada[1,2,3,4], Celso Arango [1,2,3,4] & Javier González-Peñas [1,2,3,15 ✉]

Previous research suggests an association of loneliness and social isolation (LNL-ISO) with schizophrenia. Here, we demonstrate a LNL-ISO polygenic score contribution to schizophrenia risk in an independent case-control sample (N = 3,488). We then subset schizophrenia predisposing variation based on its effect on LNL-ISO. We find that genetic variation with concordant effects in both phenotypes shows significant SNP-based heritability enrichment, higher polygenic contribution in females, and positive covariance with mental disorders such as depression, anxiety, attention-deficit hyperactivity disorder, alcohol dependence, and autism. Conversely, genetic variation with discordant effects only contributes to schizophrenia risk in males and is negatively correlated with those disorders. Mendelian randomization analyses demonstrate a plausible bi-directional causal relationship between LNL-ISO and schizophrenia, with a greater effect of LNL-ISO liability on schizophrenia than vice versa. These results illustrate the genetic footprint of LNL-ISO on schizophrenia.

[1] Department of Child and Adolescent Psychiatry, Institute of Psychiatry and Mental Health, Hospital General Universitario Gregorio Marañón, Madrid, Spain. [2] Instituto de Investigación Sanitaria Gregorio Marañón (IiSGM), Madrid, Spain. [3] CIBERSAM, Centro Investigación Biomédica en Red Salud Mental, Madrid, Spain. [4] School of Medicine, Universidad Complutense, Madrid, Spain. [5] Instituto de Investigación Sanitaria (IDIS) de Santiago de Compostela, Complexo Hospitalario Universitario de Santiago de Compostela (CHUS), Servizo Galego de Saúde (SERGAS), Santiago de Compostela, Galicia, Spain. [6] Department of Evolutionary Biology, Ecology and Environmental Sciences, Faculty of Biology, University of Barcelona, Barcelona, Spain. [7] Faculty of Medicine and Health Sciences—Psychiatry, Universidad de Oviedo, ISPA, INEUROPA, Oviedo, Spain. [8] BIOARABA Health Research Institute, OSI Araba, University Hospital, University of the Basque Country, Vitoria, Spain. [9] Hospital Universitario Virgen del Rocío, Department of Psychiatry, Universidad de Sevilla, Sevilla, Spain. [10] Hospital Universitari Institut Pere Mata, IISPV, Universitat Rovira i Virgili, Reus, Spain. [11] Neurobiology Unit, Department of Cell Biology, Interdisciplinary Research Structure for Biotechnology and Biomedicine (BIOTECMED), University of Valencia, Valencia 46100, Spain. [12] Department of Genetics, University of Valencia, Campus of Burjassot, Valencia, Spain. [13] Department of Medicine, Universitat de València, Valencia, Spain. [14] Fundación Investigación Hospital Clínico de Valencia, INCLIVA, 46010 Valencia, Spain. [15] These authors contributed equally: Álvaro Andreu-Bernabeu and Javier González-Peñas. ✉email: javier.gonzalez@iisgm.com

Social relationships are critical for emotional and cognitive development in social species[1,2]. In fact, the scientific consensus is that the need to belong to a social group is a fundamental behaviour in humans[3]. Researchers have characterized both objective and perceived (i.e., loneliness) social isolation[4,5]. While the former is an objective lack of social connections (interactions, contacts or relationships), the latter refers to the subjective feeling of distress associated with a lack of meaningful relationships, regardless of the amount of social contact[6]. Although isolated people often feel lonely, isolation is not always correlated with feelings of loneliness[4–6]. However, regardless of type, both objective social isolation and loneliness are major risk factors for morbidity and mortality[6–8], as well as for the onset of mental disorders[9–14].

Most psychiatric research on loneliness and objective social isolation has associated them with depressive symptoms and major depression[14–16], but recently researchers have shown renewed interest in their association with psychosis[17–20]. Social withdrawal and isolation are described in the early stages of schizophrenia[17,21,22], recalling the classical descriptions of pre-schizophrenia related traits by Kraepelin, Bleuler, and Conrad[23–25]. Indeed, recent meta-analyses indicate that loneliness plays an important role in the onset and maintenance of psychotic symptoms[17,22,26]. Another meta-analysis also showed a consistent association of loneliness with both positive and negative psychotic-like experiences[27]. Moreover, there are studies suggesting that loneliness may increase subclinical paranoia in non-clinical populations[28]. However, the causal relationships between social isolation and schizophrenia are still unclear[17,29].

Inherited biological factors could explain, at least partially, the relationship between social isolation and schizophrenia. Available evidence supports the genetic basis of loneliness and objective social isolation[30–33]. A recent study used multi-trait GWAS (MTAG)[34], a software developed to jointly analyse different summary statistics from related traits, to assess the genetic architecture of loneliness and objective social isolation (LNL-ISO)[32]. The researchers combined three UK Biobank GWAS datasets of (i) perceived loneliness, (ii) a proxy of social support (combined frequency of family/friends visits and living alone), and (iii) ability to confide in someone close[32]. Up to 15 genome-wide significant *loci* and SNP-based heritability estimates ($h^2_{SNP} = 4.2\%$) support the contribution of common genetic variation to this social construct. This study also found a significant genetic correlation of the combined phenotype (LNL-ISO) with schizophrenia ($rg = 0.17$, $p = 3.47 \times 10^{-12}$), consistent with a previous study reporting a significant association of perceived loneliness with schizophrenia, but not with bipolar disorder[33]. Schizophrenia polygenic scores also significantly predicted loneliness in an independent population sample in another study[35], lending further support to a shared genetic aetiology between both phenotypes.

Previous studies exploring the genetic relationship between perceived and objective social isolation and schizophrenia leave several questions unanswered, including the direction of the association, the specific biological effects of shared and non-shared predisposing variants, and the effect of additional factors on this relationship, including sex. The epidemiological and clinical presentation of psychotic disorders differs between sexes[36–38] and sex also seems to affect the perception of loneliness and the psychological impact of isolation, although results have been contradictory so far[39–41].

In this work, we aim to test the hypothesis that there is a bidirectional genetic relationship between perceived and objective social isolation and schizophrenia within a systematic and comprehensive framework (see the workflow in Fig. 1). First, we analyse loneliness and social isolation (LNL-ISO) polygenic score

contribution to schizophrenia risk in an independent Spanish case-control sample (CIBERSAM case-control sample). Second, we dissect the predisposing variation to schizophrenia according to its role in LNL-ISO and analyse the polygenic risk scores, biological profiles (using brain specific functional annotations), and sex effects across each genomic partition using an SNP subsetting approach. Third, to evaluate the role of LNL-ISO in the genetic overlap between psychiatric disorders and other related traits, we study the partial correlations between schizophrenia and related phenotypes across the LNL-ISO partitions. Finally, we perform a causality analysis between LNL-ISO and schizophrenia using a two-sample Mendelian randomization approach.

## Results

**LNL-ISO polygenic score contribution to schizophrenia risk.** We calculated polygenic scores for loneliness and isolation ($PGS_{LNL-ISO}$) using the summary statistics from the combined MTAG in the UK Biobank (UKBB) study[32] based on three different traits: (i) perceived loneliness, (ii) a proxy of social support (combined frequency of family/friends visits and living alone), and (iii) ability to confide in someone close. Figure 2A shows the percentage of variance in schizophrenia risk explained by LNL-ISO ($PGS_{LNL-ISO}$) in the independent CIBERSAM case-control sample ($N_{SCZ} = 1927$; $N_{HC} = 1561$). We found that common genetic variation predisposing to LNL-ISO significantly contributed to schizophrenia risk ($R^2$ (95% CI) = 0.56% (−0.01, 1.13) at $P_{threshold} = 0.05$, $p = 1.2 \times 10^{-4}$). One standard deviation (s.d.) increase in $PGS_{LNL-ISO}$ was associated with a 15% increase in the likelihood of belonging to the schizophrenia group (OR (95% CI) = 1.15 (1.07–1.24)). In the same target sample, LNL-ISO explained more variance in schizophrenia risk than loneliness ($R^2$ (95% CI) = 0.41% (−0.08, 0.89) at $P_{threshold} = 0.05$, $p = 1.42 \times 10^{-3}$; Fig. 2A). The contribution of $PGS_{LNL-ISO}$ to schizophrenia risk was also higher than that of ability to confide ($R^2$ (95% CI) = 0.28% (−0.11, 0.67) at $P_{threshold} = 0.05$, $p = 7.4 \times 10^{-3}$) and the two measures of social support included in LNL-ISO: number of people living in household ($R^2$ (95% CI) = 0.54% (−0.02, 1.11) at $P_{threshold} = 0.01$, $p = 3.14 \times 10^{-4}$) and frequency of family/friends visits ($R^2$ (95% CI) = 0.42% (−0.08, 0.91) at $P_{threshold} = 1$, $p = 1.2 \times 10^{-3}$; see Supplementary Data 1B).

**Polygenic dissection of schizophrenia by its relationship with LNL-ISO.** Since $PGS_{LNL-ISO}$ encompassing variants with $P_{LNL-ISO} > 0.05$ did not contribute to schizophrenia risk ($R^2$ (95% CI) = 0.052% (−0.09, 0.10) at $P_{threshold} > 0.05$, $p = 0.57$; Supplementary Data 1), schizophrenia summary statistics were subsetted according to their role in LNL-ISO GWAS (Supplementary Methods 4). Firstly, those variants not associated with LNL-ISO (SCZ[noLNL]; $P_{LNL-ISO} > 0.05$) were extracted. Second, variants associated with LNL-ISO (SCZ[LNL]; $P_{LNL-ISO} < 0.05$) were divided into those with a concordant sign of the allele effect in both schizophrenia and LNL-ISO (SCZ[CONC]; $Beta_{SCZ} > 0$ & $Beta_{LNL-ISO} > 0$ / $Beta_{SCZ} < 0$ & $Beta_{LNL-ISO} < 0$) and those with a discordant direction of the effect relative to schizophrenia (SCZ[DISC]; $Beta_{SCZ} > 0$ & $Beta_{LNL-ISO} < 0$/$Beta_{SCZ} < 0$ & $Beta_{LNL-ISO} > 0$; Fig. 1).

We performed $PGS_{scz}$ predictions on the same schizophrenia case-control sample for the three subsets of SNPs based on the dissection of SCZ summary data according to the role in LNL-ISO: $PGS_{scz}$ predictions from variants only contributing to SCZ ($PGS_{SCZ[noLNL]}$) and those contributing to both phenotypes with concordant ($PGS_{SCZ[CONC]}$) and discordant ($PGS_{SCZ[DISC]}$) effects (see Methods). Figure 2B shows the percentage of variance in schizophrenia risk explained by $PGS_{scz}$ within each subset of SNPs. $PGS_{SCZ[CONC]}$ explained almost four times more

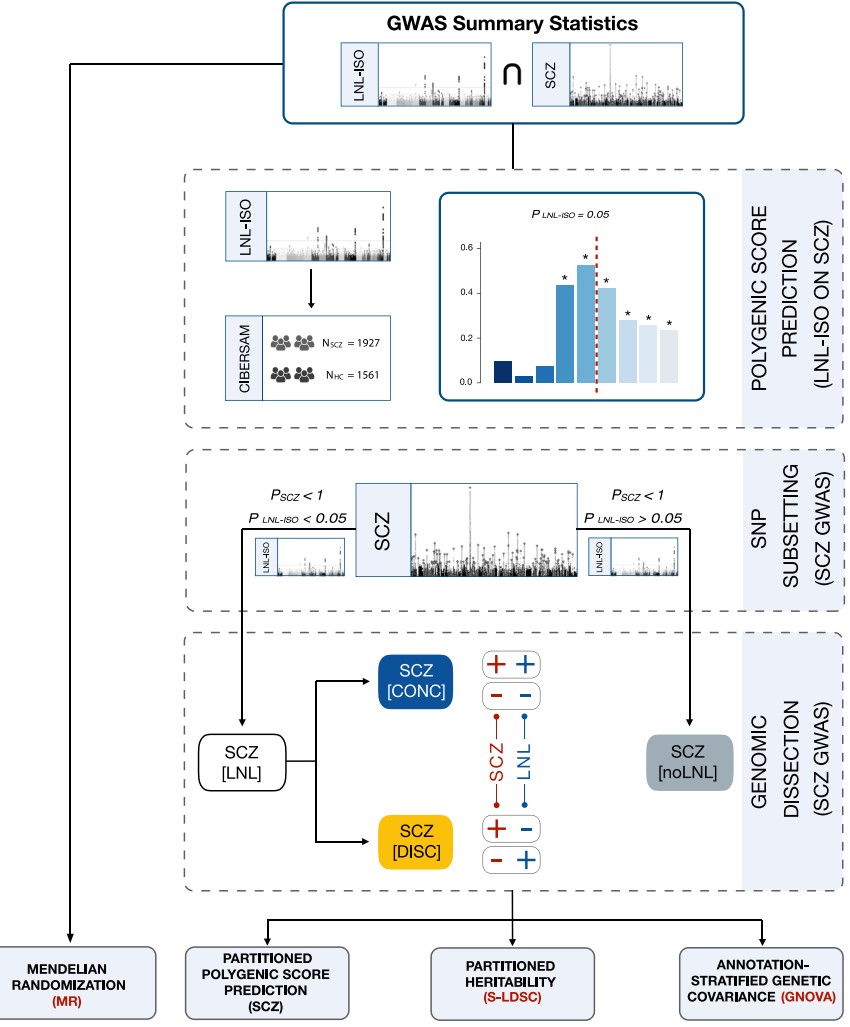

**Fig. 1 Workflow of the analytic pipeline.** GWAS summary statistics from schizophrenia[83] and LNL-ISO[32] were used. We evaluated the LNL-ISO polygenic score (PGS$_{LNL-ISO}$) contribution to schizophrenia risk in an independent case-control sample ($N_{SCZ}$ = 1927; $N_{HC}$ = 1561). Subsequent genomic dissection of schizophrenia GWAS based on LNL-ISO led to different annotations: (i) SCZ[LNL]: variants from the schizophrenia GWAS associated with LNL-ISO ($P_{LNL-ISO}$ < 0.05), (ii) SCZ[CONC]: variants from the schizophrenia GWAS associated with LNL-ISO ($P_{LNL-ISO}$ < 0.05) and concordant allele effects in both phenotypes (Beta$_{SCZ}$ > 0 & Beta$_{LNL-ISO}$ > 0 OR Beta$_{SCZ}$ < 0 & Beta$_{LNL-ISO}$ < 0), and (iii) SCZ[DISC]: variants from the schizophrenia GWAS associated with LNL-ISO ($P_{LNL-ISO}$ < 0.05) and discordant allele effects in both phenotypes (Beta$_{SCZ}$ > 0 & Beta$_{LNL-ISO}$ < 0 OR Beta$_{SCZ}$ < 0 & Beta$_{LNL-ISO}$ > 0), and (iv) (SCZ[noLNL]: variants from the schizophrenia GWAS not associated with LNL-ISO ($P_{LNL-ISO}$ > 0.05); see Methods and Supplementary Methods for further details). We performed PGS analyses, partitioned heritability, and annotation-based stratified genetic covariance analyses across those subsets. We performed Mendelian randomization to evaluate causality between schizophrenia and LNL-ISO (and its constituent traits). "+" and "−" in the figure refer to the direction of the effect of the alleles studied.

variance ($R^2$ = 3.94% at $P_{threshold}$ = 0.5, $p$ = 8.36 × 10$^{-25}$) than PGS$_{SCZ[DISC]}$ ($R^2$ = 1.02% at $P_{threshold}$ = 0.01, $p$ = 8.43 × 10$^{-8}$). PGS$_{SCZ}$ comparisons across ranked deciles were also performed (Fig. 2C, Supplementary Data 2). Higher PGS$_{SCZ}$ was found to be associated with SCZ risk across all described partitions (Fig. 2C)

Heritability estimates by LD-score regression (LDSR) found that variation within SCZ[CONC] showed a significant SNP-based heritability ($h^2_{SNP}$) enrichment, with 3.8% of the SNPs explaining an estimated 13.1% of the $h^2_{SNP}$ (Enrichment(CI$_{95\%}$) = 3.43 (2.86–4.01); $p$ = 1.83 × 10$^{-15}$; Fig. 2D; Supplementary Data 3). We found no significant heritability enrichment for SCZ[DISC] (Enrichment(CI$_{95\%}$) = 1.08 (0.58–1.58); $p$ = 0.748). Enrichment comparison of the same number of variants from SCZ[CONC] and SCZ[DISC] reflected a clear superior enrichment of concordant variants (Supplementary Fig. 2). By contrast, variants within SCZ[noLNL] harboured 65% of the SNPs and accounted for around 53.9% of the heritability,

with a relative $h^2_{SNP}$ decrease for this annotation (Enrichment(CI$_{95\%}$) = 0.81 (0.72–0.90); $p$ = 8.12 × 10$^{-5}$; Fig. 2D).

We applied partitioned heritability and LD-score regression analyses of specifically expressed genes (LDSC-SEG) within the described annotations. We observed comparable heritability enrichment profiles for SCZ[noLNL] and SCZ[CONC] across the central nervous system (CNS) and the neuronal cell type (Supplementary Data 3). The 13 brain tissues analysed displayed distinct enrichment patterns. Schizophrenia predisposing variation within SCZ[noLNL] was specifically enriched in GTEx brain cortex ($p$ = 8.5 × 10$^{-4}$) and anterior cingulate cortex ($p$ = 5.16 × 10$^{-3}$; Supplementary Fig. 3), while predisposing variation within SCZ[CONC] was enriched in GTEx hippocampus ($p$ = 0.041), although the latter was not significant after FDR correction.

We assessed PGS$_{SCZ}$ contribution to schizophrenia risk stratified by sex in the CIBERSAM case-control sample based on variation

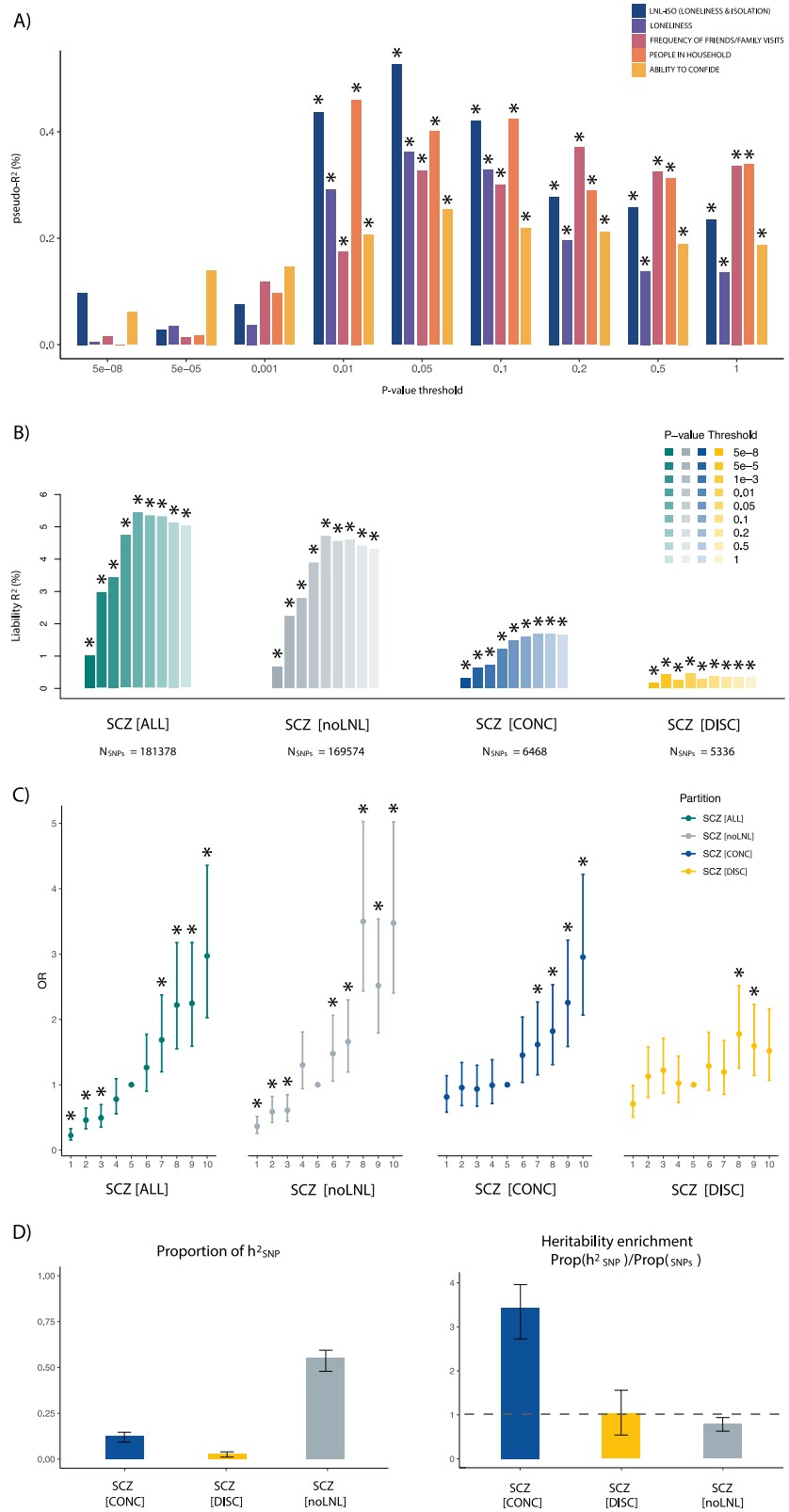

within SCZ[noLNL], SCZ[LNL], SCZ[CONC], and SCZ[DISC] (Supplementary Data 4). PGS$_{SCZ[CONC]}$ explained significantly more variance in schizophrenia risk in females ($R^2$ (95% CI) = 2.24% (1.09, 3.38) at $P_{threshold} = 0.1$, $p = 1.88 \times 10^{-13}$) than in males ($R^2$ (95% CI) = 1.41% (0.60, 2.22) at $P_{threshold} = 0.5$, $p = 2 \times 10^{-13}$), while the opposite pattern was observed for the rest of the partitions (Supplementary Data 4). We statistically

confirmed these sex-based differences using a bootstrap resampling approach comparing prediction in males and females for each genomic partition (Fig. 3, Supplementary Fig. 4).

**Annotation-stratified genetic covariance between SCZ and related phenotypes based on LNL-ISO.** We assessed covariance

**Fig. 2 Polygenic score contribution of LNL-ISO (PGS$_{LNL-ISO}$) and schizophrenia (PGS$_{SCZ}$) to schizophrenia risk and heritability estimates. A** PGS predictions of LNL-ISO (PGS$_{LNL-ISO}$) and its constituent phenotypes (see legend) on an independent schizophrenia case-control sample ($N_{SCZ}$ = 1927; $N_{HC}$ = 1561). Explained variance attributable to PGS was calculated as the increase in Nagelkerke's pseudo-$R^2$ between a linear model with and without the PGS variable. $P$-values were obtained from the binomial logistic regression of SCZ phenotype on PGS, accounting for Linkage Disequilibrium (LD) and including sex, age, and ten multidimensional scalings (MDS) ancestry components as covariates. Significant PGS predictions after FDR correction ($p_{FDR}$ < 0.05) are marked with an asterisk. See Supplementary Fig. 1 for $R^2$ values for PGS predictions on the liability scale estimated using UK Biobank prevalence for LNL-ISO constituent phenotypes. For a full detailed description and results see Supplementary Methods 3 and Supplementary Data 1. **B** PGS predictions of schizophrenia (PGS$_{SCZ}$) on an independent schizophrenia case-control sample ($N_{SCZ}$ = 1927; $N_{HC}$ = 1561). We used schizophrenia GWAS summary statistics overlapping with LNL-ISO summary statistics (SCZ(ALL)) and three subsets of them based on their effects on LNL-ISO: variants not associated with LNL-ISO (SCZ[noLNL]) and those associated with LNL-ISO with either concordant (SCZ[CONC]) or discordant (SCZ[DISC]) allele effects in each trait. Explained variance attributable to PGS was calculated as the increase in Nagelkerke's pseudo-$R^2$ between a linear model with and without the PGS variable. Pseudo-$R^2$ was converted to liability scale following the procedure proposed by Lee et al.[85] assuming a prevalence of schizophrenia in the general population of 1%[86]. $P$-values were obtained from the binomial logistic regression of SCZ phenotype on PGS, accounting for LD and including sex, age, and ten MDS ancestry components as covariates. Significant PGS predictions after FDR correction ($p_{FDR}$ < 0.05) are marked with an asterisk. For a full detailed description and results see Supplementary Methods 4 and Supplementary Data 2 A. **C** Quantile plot of PGS$_{SCZ}$ predictions from the partitions described in **B**. The target sample is separated into deciles of increasing PGS$_{SCZ}$. The case-control status of each decile is compared to the median (5$^{th}$ decile), one by one, using a logistic regression model with covariates (sex, age, and ten MDS ancestry components). OR values for each comparison were estimated from regression coefficients of these decile-status predictors. Significant comparisons ($p_{FDR}$ < 0.05) are marked with an asterisk. For a full detailed description and results see Supplementary Methods 4 and Supplementary Data 2B. **D** Proportion of SNP-based heritability ($h^2_{SNP}$) and heritability enrichment ($h^2_{SNP}/N_{SNP}$) of the annotations in schizophrenia were estimated by LD-score regression (LDSR). 95% confidence intervals based on standard errors are shown for each estimate (estimation +/− 1.96*SE). $p$-values and standard errors were calculated using a block jackknife procedure. See Supplementary Data 3 for the significance of each enrichment estimate.

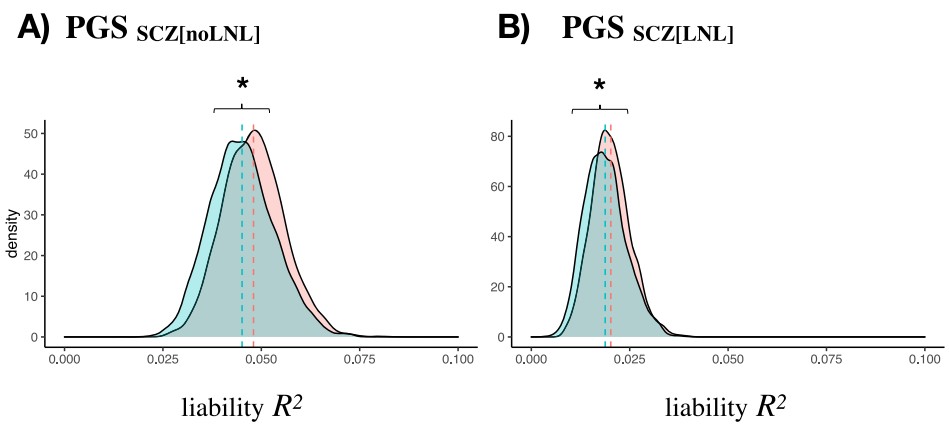

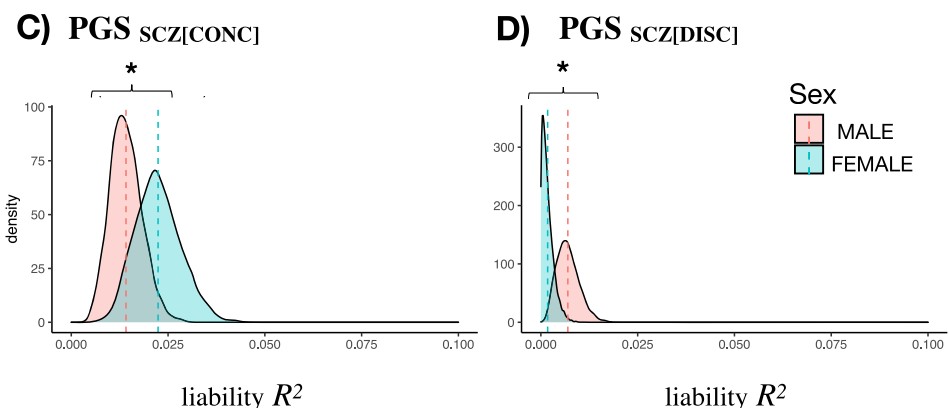

**Fig. 3 Density plot for sex comparison of PGS$_{SCZ}$ contributions to schizophrenia risk.** PGS$_{SCZ}$ predictions in case-control subsamples after bootstrap resampling (5000 permutations) of 500 schizophrenia patients (SCZ) and 500 healthy controls (HC) (selected from the overall CIBERSAM case-control sample) were performed in males ($N_{SCZ}$ = 1253; $N_{HC}$ = 859) and females ($N_{SCZ}$ = 674; $N_{HC}$ = 702), separately. Mean SCZ-HC variance explained by PGS$_{SCZ}$ on the liability scale (estimated prevalence of 0.01) in males and females was compared for predisposing variation within genome partitions. Variance explained in females and males was statistically compared with two-sided t-tests and is marked with an asterisk when it is significantly different ($p$ < 0.05). **A** PGS$_{SCZ}$ predictions comparison from variants within SCZ[noLNL]. **B** PGS$_{SCZ}$ predictions comparison from variants within SCZ[LNL]. **C** PGS$_{SCZ}$ predictions comparison from variants within SCZ[CONC]. **D** PGS$_{SCZ}$ predictions comparison from variants within SCZ[DISC].

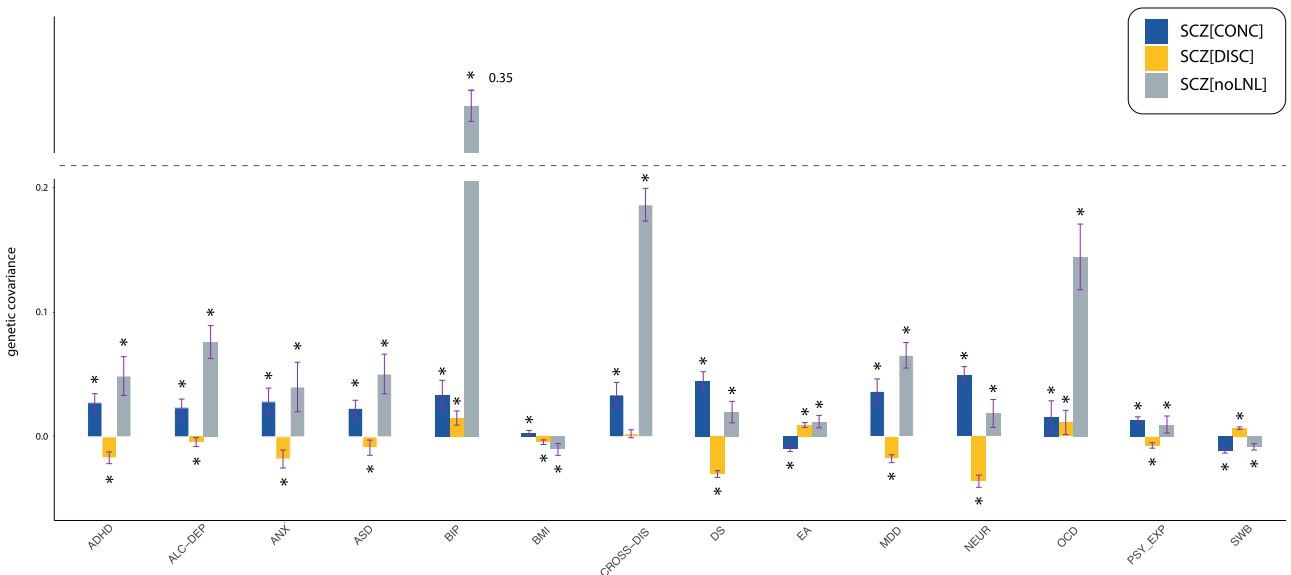

**Fig. 4 Annotation-stratified genetic covariance between schizophrenia and related traits.** We calculated covariances with GNOVA within SNP subsets from SCZ[noLNL], SCZ[CONC], and SCZ[DISC] annotations. *P*-values were calculated for the genetic covariance based on two-sided Wald tests. Error bars represent 95% confidence intervals based on standard errors (covariance estimation $+/-1.96*$SE). FDR-corrected significant associations ($p_{FDR} < 0.05$) are marked with an asterisk. Traits and disorders are abbreviated as follows: major depression (MDD), attention and deficit hyperactivity disorder (ADHD), autism spectrum disorder (ASD), anxiety disorder (ANX), bipolar disorder (BIP), obsessive-compulsive disorder (OCD), alcohol dependence disorder (ALC-DEP), cross-disorder phenotype (CROSS-DIS), neuroticism (NEUR), depressive symptoms (DS), subjective well-being (SWB), psychotic experiences in the general population (PSY-EXP), educational attainment (EA), and body mass index (BMI). For further details of the phenotypes see Supplementary Data 6 and Supplementary Methods 6.

between predisposing genetic variation to schizophrenia and a series of neuropsychiatric disorders and related phenotypes across SCZ[noLNL], SCZ[CONC], and SCZ[DISC] using GNOVA[42].

The majority of the disorders (MDD, ANX, ADHD, ASD, CROSS-DIS, ALC-DEP) and personality traits (NEUR, SWB, DS, PSY_EXP) tested here showed positive genetic correlation within SCZ[CONC] and negative genetic correlation within SCZ[DISC] (Fig. 4). However, BIP and OCD showed positive covariances within both genomic annotations. Therefore, alleles that increase the risk for SCZ but decrease the risk for LNL-ISO (SCZ[DISC]) are positively correlated with BIP or OCD, but negatively correlated with MDD, ASD, ADHD or ANX, providing one distinction between these two groups in their relationship with LNL-ISO. As expected, estimated correlations within SCZ[noLNL] were similar to those previously described for schizophrenia across the whole genome[43] (Fig. 4, Supplementary Data 5).

**Mendelian randomization.** We finally assessed the direction of causation between social isolation (LNL-ISO) and schizophrenia using a range of bidirectional Mendelian randomization methods (Inverse Variance Weighted (IVW)[44], Weighted Median (WM)[45], MR-Egger[46], Simple Mode (SM)[47], Weighted Mode (WM)[47], MR-PRESSO[48] and CAUSE[49]). We used multiple tests to rule out horizontal pleiotropy (Table 1 and Supplementary Data 7).

We found evidence for a strong bidirectional causal effect of LNL-ISO on schizophrenia (IVW$-\beta$ (standard error (SE)) = 1.11(0.48), $p = 0.021$), (WM$-\beta$ (standard error (SE)) = 1.37 (0.40), $p = 6.14 \times 10^{-4}$) with a consistent direction of the effects across methods except in the case of MR-Egger. Although we did not detect horizontal pleiotropy with the MR-Egger intercept analysis ($p = 0.36$), there was evidence of heterogeneity (IVW Q-p-value = $2.94 \times 10^{-6}$) (Table 1, Supplementary Fig. 5). In this scenario, the WM method, which is more robust in the presence

of outliers, was preferred over the IVW method[45,50]. The presence of heterogeneity also provided the most suitable explanation for the difference in the direction of Egger´s effect due to the sensitivity of this method to the presence of outliers and heterogeneity, which lead to poor causal estimates in such situations[46] (see Table 1 and Supplementary Methods 7).

Additional robust methods that eliminate outliers that may be influencing the outcome due to pleiotropy (MR-PRESSO[48]) or account for both correlated and uncorrelated pleiotropy (CAUSE[49]) showed comparable results to those using the WM method, with even larger effect sizes using MR-PRESSO (MR-PRESSO outlier-correction β (Sd) = 1.45(0.30), $p = 0.001$; CAUSE$-\gamma$ (CI95%) = 0.61 (0.34, 0.89), $p = 0.003$) (Table 1).

We also found a causal effect of schizophrenia liability on LNL-ISO (WM$-\beta$ (SE) = 0.015(0.005), $p = 0.008$; CAUSE$-\gamma$ (CI95%) = 0.01 (0.01, 0.01), $p = 0.003$), with evidence of heterogeneity (IVW Q-p-value: $2.21 \times 10^{-11}$) but no indication of horizontal pleiotropy based on the MR-Egger intercept analysis ($p = 0.48$).

In the MR analyses including the constituent phenotypes of LNL-ISO, we found comparable evidence for bidirectional causality between perceived loneliness and schizophrenia to that found for LNL-ISO (Table 1, Supplementary Figs. 5 and 6). We also found a unidirectional negative causal effect of ability to confide on schizophrenia (WM$-\beta$ (SE) = $-0.6$ (0.19), $p = 0.002$), and a unidirectional negative causal effect of schizophrenia on number of people in household (WM$-\beta$ (SE) = $-0.011$ (0.003), $p = 5.86 \times 10^{-3}$). We found no evidence of causality between the number of family/friends visits and schizophrenia.

## Discussion

This work suggests the presence of genetic overlap between social isolation, measured using LNL-ISO, and schizophrenia, with a bidirectional causal relationship. We found that overlapping predisposing genetic variation with concordant effects in both

**Table 1 Bidirectional causal inference analyses between loneliness and isolation phenotypes and schizophrenia.**

Causal Effects of Loneliness and Isolation related traits on Schizophrenia

| Exposure | Number of Instruments | Outcome | IVW β (SE) | IVW P-value | Weighted Median β (SE) | Weighted Median P-value | MR-Egger β (SE) | MR-Egger P-value | MR-Egger Intercept P-value | Heterogeneity Q-P-Value | MR-PRESSO Outliers (n) | MR-PRESSO β (Sd) | MR-PRESSO P-value | CAUSE γ (CI95%) | CAUSE P |
|---|---|---|---|---|---|---|---|---|---|---|---|---|---|---|---|
| **LNL-ISO MTAG** | 13 | Schizophrenia | 1.114 (0.48) | 0.021 | **1.372 (0.40)** | **6.14E-04** | −1.112 (2.36) | 0.646 | 0.36 | 2.94E-06 | 3 | **1.453 (0.30)** | **0.001** | **0.61 (0.34, 0.89)** | **0.003** |
| **Loneliness UKBB** | 14 | Schizophrenia | 1.366 (123) | 0.260 | **2.640 (0.94)** | **0.005** | 3.797 (6.01) | 0.538 | 0.68 | 8.00E-07 | 2 | **2.785 (0.75)** | **0.004** | **1.32 (0.62, 2.02)** | **0.014** |
| Friends/Family visits | 19 | Schizophrenia | 0.486 (0.31) | 0.13 | 0.446 (0.25) | 0.08 | 2.511 (1.69) | 0.156 | 0.24 | 2.20E-06 | 2 | 0.524 (0.19) | 0.015 | 0.21 (0.01, 0.43) | 0.210 |
| Number in household* | 15 | Schizophrenia | −0.699 (0.71) | 0.32 | −0.174 (0.57) | 0.76 | −0.619 (3.03) | 0.841 | 0.97 | 1.6E-06 | 2 | −0.574 (0.48) | 0.25 | −1.09 (−2.19, 0.01) | 0.220 |
| **Able to confide** | 12 | Schizophrenia | **−0.718 (0.21)** | **0.001** | **−0.606 (0.19)** | **0.002** | −0.667 (1.09) | 0.554 | 0.96 | 3.50E-04 | 2 | **−0.683 (0.14)** | **0.001** | −0.19 (−0.33, −0.05) | 0.065 |

Causal Effects of Schizophrenia on Loneliness and Isolation related traits

| Exposure | Number of Instruments | Outcome | IVW β (SE) | IVW P-value | Weighted Median β (SE) | Weighted Median P-value | MR-Egger β (SE) | MR-Egger P-value | MR-Egger Intercept P-value | Heterogeneity Q-P-Value | MR-PRESSO Outliers (n) | MR-PRESSO β (Sd) | MR-PRESSO P-value | CAUSE γ (CI95%) | CAUSE P |
|---|---|---|---|---|---|---|---|---|---|---|---|---|---|---|---|
| Schizophrenia | 69 | LNL-ISO MTAG | 0.012 (0.005) | 0.020 | **0.015 (0.005)** | **0.008** | 0.026 (0.02) | 0.190 | 0.48 | 2.21E-11 | 2 | **0.012 (0.004)** | **0.013** | **0.01 (0.01, 0.01)** | **0.003** |
| Schizophrenia | 69 | Loneliness UKBB | 0.004 (0.002) | 0.067 | **0.007 (0.002)** | **2.09E-04** | 0.010 (0.008) | 0.230 | 0.47 | 6.92E-16 | 3 | 0.004 (0.002) | 0.025 | **0 (0, 0.01)** | **0.002** |
| Schizophrenia | 69 | Friends/Family visits | 0.005 (0.007) | 0.458 | 0.007 (0.007) | 0.33 | 0.005 (0.025) | 0.826 | 0.99 | 1.53E-17 | 2 | 0.008 (0.006) | 0.165 | 0 (0, 0.01) | 0.220 |
| **Number in household** | 69 | Schizophrenia | **−0.008 (0.003)** | **0.007** | **−0.010 (0.003)** | **5.86E-03** | −0.003 (0.011) | 0.788 | 0.63 | 3.12E-03 | 2 | −0.006 (0.002) | 0.021 | −0.01 (−0.01, 0) | 0.025 |
| Schizophrenia | 69 | Able to confide | −0.012 (0.009) | 0.184 | −0.017 (0.03) | 0.09 | −0.014 (0.03) | 0.68 | 0.95 | 1.53E-05 | NA | −0.012 (0.009) | 0.188 | −0.01 (−0.02, 0) | 0.032 |

Traits in bold have significant results in at least one method except for "Number of people in household" due to a lower number of pleiotropic variants Variables at $p < 5 \times 10^{-8}$ for all the traits except for "Number of people in household" due to a lower number of genome-wide significant SNPs at that threshold. For this trait, we used a threshold of $p < 5 \times 10^{-6}$ instead. The column "Outliers" reports the number of pleiotropic variants removed with MR-PRESSO. MR-PRESSO β-Effects were estimated after removing the outliers. IVW, inverse variance weighted linear regression, SE, Standard error measure of effect size, Q-P-Value, P-value of IVW Cochran's Q statistic. LNL-ISO MTAG, Multi-trait GWAS of loneliness and social isolation. Loneliness UKBB, loneliness trait from the UK Biobank. Friends/family visit, UK Biobank trait of frequency of friends/family visits. Able to confide, UK Biobank trait of frequency of confide in someone close to you. γ (CI95%) Posterior median and 95% credible intervals of the true value of causal effect with CAUSE. P (CAUSE): p-value testing that causal model is better than sharing model using ELPD test (significance level $p < 0.05$).

phenotypes shows significant SNP-based heritability enrichment, supporting the relatively enhanced contribution of this set of variants to schizophrenia liability. We found the concordant variation to contribute more to schizophrenia risk in females and to be positively correlated with other neuropsychiatric traits. Conversely, discordant variation contributed to schizophrenia risk only in males and was negatively correlated with most neuropsychiatric traits. These results reveal the likely genomic footprint of social isolation on the heritability of schizophrenia and provide new insights about their relationship[32,35]. They also support the role of LNL-ISO as a critical social trait for understanding the heterogeneity of pleiotropic genetic effects between schizophrenia and other psychiatric disorders and behavioural traits. In fact, each of the individual traits included in the composite LNL-ISO phenotype had a significant polygenic score contribution to schizophrenia risk. These results agree with separate findings of a clinical overlap between schizophrenia and both perceived loneliness and objective social disconnection and support the idea that social isolation may play a significant role in the aetiology of psychotic disorders[17,19,20,27].

Researchers have described polygenic score predictions and LD-score-based partition heritability estimates as powerful methods for evaluating the effects of genetic predisposing variation within specific subsets of variants[51–53]. With 3.8% of SNPs explaining an estimated 13.1% of SNP-based heritability, concordant overlapping variation between both phenotypes exhibits more than a three-fold increase in heritability enrichment compared to variants not predisposing to LNL-ISO, and a much higher enrichment than most of the genome-wide annotations previously evaluated in schizophrenia[54]. LDSC-SEG analyses pointed to a significant enrichment at the uncorrected level of concordant overlapping variation in the hippocampus, a brain region involved in social behaviour[55,56] and cognitive flexibility[57], which may be especially sensitive to brain inflammation caused by loneliness and isolation[10,56]. In this respect, recent work has described loneliness affecting the white matter integrity of the hippocampus[58].

Despite reported sex differences in the epidemiology and clinical manifestations of psychotic disorders[36,38,59], previous studies had not found an effect of sex on genetic associations[60]. By analysing the genomic overlap between schizophrenia and LNL-ISO, we did observe a differential effect of sex on polygenic contributions to schizophrenia risk. Concordant overlapping variants in SCZ and LNL-ISO accounted for a significantly greater amount of variance in schizophrenia risk in females than in males, while the opposite pattern was observed in the rest of LNL-ISO based annotations. These results are in line with recent studies suggesting a potentially higher impact of loneliness and objective social isolation on psychiatric outcomes in females than in males[26,41]. This may be due to a more negative perception of social deprivation in females related to their role in modern society[61] and a greater protective effect of an enriched social network in males[62]. Moreover, among patients with schizophrenia, loneliness has been described to be more prevalent in females than males[63]. Our results suggest the existence of a social-related environment differentially affecting males and females that could be, at least in part, responsible for the different sex-stratified PGS contributions. Further studies should evaluate the impact of sex and gender differences in subjective social perception in epidemiological models.

Genetic correlations have been shown to be a very useful method for understanding shared genetic architecture and the interrelationship between disorders and related traits, despite some limitations[43,64–66]. By evaluating annotation-stratified correlations, previous studies have described subtle structures in shared genetics between complex traits[42,67,68]. In our study, we

have described the impact of the genetic liability to LNL-ISO in the relationship between schizophrenia and most of the tested neuropsychiatric disorders (ASD, MDD, ANX, ADHD, ALC-DEP) and other related behavioural traits (SWB, NEUR, PSY-EXP, EA). In the majority of the disorders, schizophrenia is positively correlated within concordant overlapping variation and negatively correlated within discordant overlapping variation with LNL-ISO, thus pointing to a shared genetic impact of social isolation on comorbidity with these disorders. However, OCD and BIP have positively correlated with schizophrenia regardless of LNL-ISO based annotations, thus suggesting that the association of these disorders with schizophrenia is independent of the genetic predisposition to LNL-ISO. These results are in line with recent findings suggesting that schizophrenia, BIP, and OCD could belong to the same psychopathology factor at the genomic level[69].

The genetic relationship between schizophrenia with EA and other cognitive-related measures such as intelligence test performance has been widely studied[68,70,71]. Assessing annotation-stratified genetic covariance between EA and schizophrenia, we described a negative covariance within concordant overlapping variation, while EA showed a positive correlation with schizophrenia across discordant overlapping variation and with variants only associated with schizophrenia. Our findings suggest that poor educational attainment often found in young patients with schizophrenia[72,73] could be mediated by social isolation.

Mendelian randomization analyses provided evidence of the bidirectional nature of the causal relationship between loneliness and isolation and schizophrenia liability, with greater size of the effect of LNL-ISO on schizophrenia risk than in the opposite direction. This finding of bidirectional causality between social isolation and schizophrenia was confirmed with the recently developed method CAUSE, which provides better control for correlated and uncorrelated horizontal pleiotropy[49]. Our results are consistent with previous evidence suggesting that loneliness and objective social isolation could trigger both positive and negative psychotic symptoms in clinical and non-clinical populations[17,27]. It could also explain the high levels of loneliness and isolation before the onset of psychosis in individuals at clinical high risk for psychosis[74]. On the other hand, the described effect of schizophrenia liability on social isolation could also give an explanation to the high prevalence of loneliness in the chronic stages of psychotic illnesses[17,20,26].

Causal inferences assessing the relationships between LNL-ISO constituents and schizophrenia also found a unidirectional negative causal effect of "ability to confide" on schizophrenia, in line with recent studies describing the association of lack of confidence and loneliness with psychosis, which may be mediated by negative schemata of others[29,75]. Moreover, a unidirectional negative causal effect of schizophrenia liability on the "number of people living in your household" was found, thus suggesting a possible indirect causal effect of schizophrenia genetic liability on subsequent social disconnection in participants diagnosed with schizophrenia[18]. This relationship is also reinforced with the significant polygenic contribution of both phenotypes to schizophrenia risk (Fig. 2A).

Previous studies assessing social determinants of poor mental health have evaluated the association of social disadvantage and their genetic determinants with the risk of psychosis[76,77]. Our study adds to this previous evidence by incorporating a subjective perception to social dysfunction in psychosis from a genetic perspective. Further studies should explore the effect of subjective perception of loneliness and its association with the social defeat hypothesis with the risk of psychosis[76].

Our study was subject to several limitations. First, we used measures of loneliness and objective social isolation from the UKBB, which are based on single-question questionnaires and not on validated scales such as UCLA loneliness[78]. Nevertheless, multiple research studies have previously validated binary self-reported loneliness questionnaires and found strong convergent validity with UCLA loneliness scale[1,58,78]. Second, since we used discovery samples for polygenic score analysis from the UKBB population, socio-economic biases could have affected our genetic predictions to some extent[79,80]. Third, partitioning the genome in order to estimate heritability enrichment in a reduced subset of SNP may have underpowered some of our analyses. Larger sample sizes in future studies could address this limitation. Fourth, we found a great degree of heterogeneity in the MR analyses. However, we implemented several complementary methods to support the robustness of our findings and report only on results that held up across all methods. Other methods for genomic dissection such as Genomic SEM[81] could be used in future studies to strengthen the results presented here. Finally, the small effect sizes suggest that even if genetic variation may partially underpin the link between schizophrenia and LNL-ISO phenotypes, environmental variables are likely to play a substantial role in this association and should be explored in future epidemiological studies.

In summary, our results shed additional light on the relationship between social isolation and schizophrenia from a genetic perspective, and lend further support for the potential role of LNL-ISO in the onset and maintenance of schizophrenia and other psychotic disorders[82]. We also provide new insights into the influence of social isolation on comorbidity with other mental disorders and its interplay with behavioural traits. Given that social isolation and perceived loneliness may be modifiable, they could be targets for effective preventive interventions with a potentially substantial impact on mental health.

## Methods

**Samples and GWAS summary data**. We used a case-control sample including 1927 schizophrenia cases (65% males) and 1,561 healthy controls (HC) (55% males) from CIBERSAM (Centro de Investigación Biomédica en Red en Salud Mental, Spain) as an independent target sample for polygenic score predictions (SCZ_CIBERSAM). All participants were genotyped as part of the Psychiatric Genomics Consortium (PGC), and passed quality control (QC filters) per PGC-SZ2 criteria[83]. See Supplementary Methods for a detailed description. Informed consent signed by each participating subject or legal guardian and approval from the corresponding Research Ethics Committee were obtained before starting the study.

We used the following genetic summary statistics from the previous GWAS: (i) schizophrenia GWAS from the Psychiatric Genetic Consortium (PGC-SCZ2) comprising 35,476 cases and 46,839 controls[83], (ii)LNL-ISO combined phenotype (LNL-ISO)[32] GWAS based on the combined multi-trait GWAS (MTAG) in the UKBB study, yielding an effective maximum sample size of 487,647 individuals, and (iii) the latest UKBB GWAS results for the independent loneliness and isolation traits that were included in the original LNL-ISO MTAG: (a) loneliness UKBB, (b) a proxy of social support, as measured by the frequency of family and friend visits and the number of people living in household, and (c) ability to confide in someone close to you. There is no overlap between PGC-SCZ2 and SCZ_CIBERSAM samples. Another recent schizophrenia GWAS[84], including approximately 5000 new cases and 18,000 controls to PGC2, was also used to rule out changes in risk predictions or heritability estimates compared to PGC-SCZ2 GWAS (Supplementary Data 2).

**Dissection of schizophrenia summary genetic data based on LNL-ISO**. First, we selected variants that were included in both schizophrenia and LNL-ISO summary data. Second, we divided schizophrenia summary statistics from the set of overlapping variants ($N\_SNPs = 5,658,282$) into two different subsets of variants, according to their effects in the LNL-ISO (Fig. 1): those variants not associated with LNL-ISO (SCZ[noLNL]; $P_{LNL-ISO} > 0.05$; $N\_SNPs = 5,172,017$) and those variants associated with LNL-ISO (SCZ[LNL]; $P_{LNL-ISO} < 0.05$; $N\_SNPs = 486,265$). We selected this cutoff based on the fact that LNL-ISO-based PGS ($PGS_{LNL-ISO}$) prediction on schizophrenia in the case-control target sample from CIBERSAM at $P_{LNL-ISO} > 0.05$ was not significant ($R^2$ (CI95%) = 0.052% (−0,09,0,19) at $P_{threshold} > 0.05$, $P = 0.569$; Supplementary Data 1C). Third, based on the concordance or discordance of the effects of the same effect allele, we again divided SCZ[LNL] into those variants with concordant (SCZ[CONC]; $N\_SNPs = 269,361$) or discordant (SCZ[DISC]; $N\_SNPs = 216,904$)

effects in schizophrenia and LNL-ISO (SCZ[CONC]: $Beta_{SCZ} > 0$ & $Beta_{LNL-ISO} > 0$ OR $Beta_{SCZ} < 0$ & $Beta_{LNL-ISO} < 0$; SCZ[DISC]: $Beta_{SCZ} > 0$ & $Beta_{LNL-ISO} < 0$ OR $Beta_{SCZ} < 0$ & $Beta_{LNL-ISO} > 0$). In each of the final datasets, we removed correlated SNPs due to linkage disequilibrium (LD) using PLINK 1.9 clumping algorithm ($r^2 > 0.1$; window size = 500 kb). See Supplementary Methods 4 for further details.

**Polygenic score (PGS) models**. We performed polygenic models based on PGC-SZ2[83] ($PGS_{SCZ}$) and LNL-ISO[32] ($PGS_{LNL-ISO}$) GWAS summary statistics as the discovery samples, and SCZ_CIBERSAM case-control sample as the target sample (N_SCZ = 1927; N_HC = 1561). Several P thresholds were used ($P < 5 \times 10^{-8}$, $5 \times 10^{-5}$, $1 \times 10^{-3}$, 0.01, 0.05, 0.1, 0.2, 0.5 and 1). Genetic variants within the Major Histocompatibility Complex (MHC) were removed. We calculated standardized PGS and evaluated significance by logistic regression, using case-control status as dependent variable and sex, age, and ten first multidimensional scaling (MDS) ancestry components as covariates. Explained variance attributable to PGS was calculated as the increase in Nagelkerke's pseudo-$R^2$ between a model with and without the PGS variable. In PGS predictions with PGC-SZ2, Nagelkerke's pseudo-$R^2$ were converted to liability scale following the procedure proposed by Lee et al.[85] assuming a prevalence of schizophrenia in the general population ~1%[86]. We applied a correction for multiple testing to all p-values. CI for the increase in $R^2$ was estimated through bootstrap resampling (N = 5000 permutations), applying the Normal Interval method, after checking the normality of the bootstrap distribution. In order to compare $PGS_{LNL-ISO}$ predictions with LNL-ISO's constituent phenotypes (loneliness UKBB, frequency of family and friend visits, number of people living in household, and ability to confide), PGS contributions of these phenotypes to schizophrenia risk were also evaluated in the same target sample.

Using the described separated subsets of variants based on their effect in LNL-ISO, we also calculated $PGS_{SCZ}$. LD-independent variants within SCZ[noLNL] (N_clumped SNPs = 169,574), SCZ[LNL] (N_clumped SNPs = 11,804), SCZ[CONC] (N_clumped SNPs = 6468) and SCZ[DISC] (N_clumped SNPs = 5,336) were used to calculate PGS on the SCZ_CIBERSAM case-control sample. We calculated standardized PGS and evaluated significance with logistic regression models as described above.

In order to assess the effect of sex on these models, we compared the explained variance in the case-control status for predisposing variation to schizophrenia within SCZ[noLNL], SCZ[LNL], SCZ[CONC] and SCZ[DISC] in females and males. Then, after bootstrap resampling (5000 permutations) of 500 schizophrenia and 500 HC subjects in each sex separately (see Supplementary Methods 3), we statistically compared the differences between the distribution of liability $R^2$ in males and females across each genomic partition with two-sided t-tests. Since no sex differences have been reported in schizophrenia overall prevalence[87,88] we considered a prevalence estimate of 1% for both sexes. We also conducted a sensitivity analysis using recent prevalence estimates in the Spanish population[89] (prevalence in males = 0.0079 and females = 0.0045), with comparable findings (Supplementary Fig. 4).

In order to understand the direction of the effect of the PGS across the different partitions (higher or lower values in SCZ patients compared to healthy controls), $PGS_{SCZ}$ comparisons across ranked deciles were also performed. The target sample was first separated into ten deciles of increasing PGS. The P-threshold with the lowest p-value was selected for each partition. The phenotype values of each decile were compared to those of the reference decile (the median (5th decile) was used as a reference) one by one, with decile status as a predictor of target phenotype (5th decile was coded 0 and tested decile 1) in a logistic regression model. OR values for each comparison were estimated from regression coefficients of these decile-status predictors. Sex, age, and ten first MDS ancestry components were used as covariates.

The term "prediction" is used in relation to polygenic score models to conform to standard terminology in the field. However, these models are not used with a predictive purpose.

**LD-score regression (LDSR) and heritability estimates**. We calculated SNP-based heritability ($h^2_{SNP}$) estimates for resulting genome partitions from dissections of schizophrenia summary genetic data based on LNL-ISO as described before: (i) SCZ[noLNL], SCZ[CONC], and SCZ[DISC] annotations; and (ii) sub-annotations from the intersection between those annotations (SCZ[noLNL] and SCZ[CONC]) and gene expression data from ten whole tissues[64], 13 brain-related tissues (Brain GTEx)[51], and 3 brain cell-type annotation files (neurons, astrocytes, and oligodendrocytes)[51,90]. The intersection with SCZ[DISC] was not included due to the low $h^2_{SNP}$ for this annotation. ldsc v1.0.1[51], a command-line tool for estimating heritability, was used. We performed both heritability enrichment analyses across the described annotations (-h2) and one-sided t-tests to evaluate whether the cell-type enrichment in schizophrenia within a particular LNL-ISO annotation was higher than the same cell-type enrichment in schizophrenia outside the LNL-ISO annotation (-h2-cts) (see Supplementary Methods 5). Additional information on the whole procedure is described in Supplementary Methods 5.

**Annotation-stratified genetic covariance**. To examine the influence of LNL-ISO based annotations (SCZ[noLNL], SCZ[CONC], and SCZ[DISC]) on the correlation between schizophrenia and other related disorders or traits, we calculated partial correlations using GNOVA[42] (https://github.com/xtonyjiang/GNOVA). First, we selected neuropsychiatric and related phenotypes reportedly showing significant correlations with schizophrenia and/or loneliness/social isolation phenotypes in previous studies (i.e., major depression (MDD), attention and deficit hyperactivity disorder (ADHD), autism spectrum disorder (ASD), anxiety disorder (ANX), bipolar disorder (BIP), obsessive-compulsive disorder (OCD), alcohol dependence disorder (ALC-DEP), cross-disorder phenotype (CROSS-DIS)—based on a meta-analysis across eight mental disorders with a total sample of 232,964 cases and 494,162 controls— neuroticism (NEUR), depressive symptoms (DS), subjective well-being (SWB), psychotic experiences in the general population (PSY-EXP), and educational attainment (EA)). We also included body-mass index (BMI) summary data since some researchers report that this phenotype may be influenced by LNL-ISO[32] (see Supplementary Methods 6 for references). We used covariance estimates based on partial correlations restricted to SNP subsets within each annotation conducted with GNOVA[42]. Derived p-values were statistically corrected using a Benjamini–Hochberg False Discovery Rate (FDR) procedure ($p_{FDR} < 0.05$). See Supplementary Methods 6 for further details.

**Two-sample Mendelian randomization**. We used Mendelian Randomization (MR) to investigate the direction of the causal relationships between social isolation (LNL-ISO) and its constituents (i.e., loneliness UKBB, frequency of family visits, number of people in household, and ability to confide) with schizophrenia liability using the latest GWAS data available in MRC-IEU API resource[91,92] (https://gwas.mrcieu.ac.uk; https://mrcieu.github.io/ieugwasr/).

We selected genome-wide significant SNPs at $p < 5 \times 10^{-8}$ except in the case of a number of people in household due to an insufficient number of instrumental variables (IV) at this threshold. We used a $p < 5 \times 10^{-6}$ instead. We applied a default LD-Clumping $r^2$ threshold of 0.001 and a window of 10,000 kb. Five MR methods: (i) Inverse variance-weighted method (IVW)[44], (ii) WM[45], (iii) MR-Egger[46], (iv) SM, and (v) WMo[47] were performed using the R package TwoSampleMR v.0.5.3[92] (https://github.com/mrcieu/TwoSampleMR). We conducted sensitivity tests including heterogeneity tests (IVW and Egger Cochran's Q statistic test)[46], leave-one-out, and pleiotropy tests using functions of the same R package. Additionally, we conducted Mendelian Randomization Pleiotropy RESidual Sum and Outlier (MR-PRESSO)[48] analyses and a novel MR latent-model method (CAUSE)[49] to further account for pleiotropy (https://jean997.github.io/cause/).

We applied a correction for multiple testing using Benjamini–Hochberg FDR ($p_{FDR} < 0.05$). See Supplementary Methods 7 for further details.

**Reporting summary**. Further information on research design is available in the Nature Research Reporting Summary linked to this article.

## Data availability
GWAS summary datasets used in this study have been downloaded from the following repositories: UK BIOBANK. https://www.nealelab.is/uk-biobank. MRBASE—IEU GWAS PROJECT. https://gwas.mrcieu.ac.uk/. PGC. https://www.med.unc.edu/pgc/download-results/. SSGAC. https://thessgac.com/. Summary statistics for the LNL-ISO composite (LNL-ISO) were downloaded from the following repository: (https://doi.org/10.17863/CAM.23511). Individual genotype data for the CIBERSAM consortium samples (Spain, https://www.cibersam.es/en) used here and analytic code is available from the corresponding author upon reasonable request, since the deposit of this data to a public repository is not allowed due to ethical and legal requirements at the participating centres. Functional and cell-type annotation files (Neuron, astrocytes and oligodendrocytes) for heritability analyses were downloaded from the public available LDSC repository (http://data.broadinstitute.org/alkesgroup/LDSCORE/).

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

## Acknowledgements

This work was supported by the Spanish Ministry of Science and Innovation. Instituto de Salud Carlos III (SAM16PE07CP1, PI16/02012, PI17/00997, PI19/01024, PI20/00721), co-financed by ERDF Funds from the European Commission, "A way of making Europe", CIBERSAM. Madrid Regional Government (B2017/BMD-3740 AGES-CM-2), European Union Structural Funds. European Union Seventh Framework Program under grant agreements FP7-4-HEALTH-2009-2.2.1-2-241909 (Project EU-GEI), FP7-HEALTH-2013-2.2.1-2-603196 (Project PSYSCAN), and FP7- HEALTH-2013-2.2.1-2-602478 (Project METSY); and European Union H2020 Program under the Innovative Medicines Initiative 2 Joint Undertaking (grant agreement No 115916, Project PRISM, and grant agreement No 777394, Project AIMS-2-TRIALS), Fundación Familia Alonso, Fundación Alicia Koplowitz, and Fundación Mutua Madrileña. We are grateful to Paloma Pantoja for her assistance in the design of the figures. A.A.-B. holds a Rio Hortega Grant from Instituto de Salud Carlos III (CM20/00114). C.M.D.-C. holds a Juan Rodés Grant from Instituto de Salud Carlos III (JR19/00024).

## Author contributions

J.G.P., A.A.-B., J.C., and C.M.D.-C. contributed to the conception and design of the study. J.G.P. designed the analytic pipeline. J.G.P. and A.A.-B. were involved in data analysis and writing the paper. C.M.D.-C., J.C., G.M., L.d.H., B.C.-F., E.V., and C.A. edited the paper. The rest of the authors (X.G., C.A., L.F., J.B., A.G.P., L.M., J.N., M.D.M., E.J.A., and M.P.) were involved in patient recruitment and data collection. All the authors have reviewed and approved the final version of the manuscript.

## Competing interests

C.A has been a consultant to or has received honoraria or grants from Acadia, Angelini, Gedeon Richter, Janssen Cilag, Lundbeck, Minerva, Otsuka, Roche, Sage, Servier, Shire, Schering Plough, Sumitomo Dainippon Pharma, Sunovion, and Takeda. B.C.-F. has received honoraria (advisory board and educational lectures) and travel expenses from Takeda, Menarini, Angelini, Teva, Otsuka, Lundbeck, and Johnson & Johnson. He has also received unrestricted research grants from Lundbeck. Xaquín Gurriarán has received a grant from Fundación Instituto Roche. C.M.D.-C. has received honoraria from AbbVie, Sanofi, and Exeltis. The rest of the authors declare no competing interests.
