## [Peer Review File · Nature Communications]

Title: Polygenic contribution to the relationship of loneliness and social isolation with schizophreniaREVIEWER COMMENTS

Reviewer #1 (Remarks to the Author):

This was an interesting paper, which performed a number of novel analyses, it was however lacking a clear thread to the argument, it was not clear how different parts of the paper related to each other, and some of the results were not fully discussed. Thus is it not always clear why the analysis is done, and what has been learnt from it. There was also a general tendency towards sentences that resembled lists and frequent acronyms (often with slightly ugly "_" in them). Together these made it hard to follow in places. A few specific comments are below:

The results section needs a more sentence to give a high level explanation rather than immediately pointing to a figure. Bearing in mind that the methods section is after the main text, some context for what is being done and why is needed.

What was the rationale for doing permutation tests for the PGS to schizophrenia? It seems unnecessary.

If you are going to talk about the relative size of an R^2 (as done in line 164) it would make sense to report the CIs.

Figure 1 - I think it would be easier to have figures that didn't use quite so many acronyms, I think it would be better to spell out the factors. I don't like the formatting in part C of having a legend in place of axis labels.

I think the idea of splitting the data into sets based on the associations with loneliness needs to be explored a little more in the main text of the results - currently it is described in the legend to Figure 1, which doesn't seem like the optimal place.

I wonder if the comparative measures of heritability in lines 197-204 is an example of winners' curse? I.e. that the SNPs that occur in the loneliness data are simply more powered. What is the estimated heritability of the top 3.8% of the SNPs in SCZ-noLNL?

Figure 2 it would be useful to see the means and CIs for this in a supplementary table.

Figure 3 More detail on what "Cross-disorder phenotype" is.

Lines 261-264, this is very list-y sentence, and it hard to follow in it's current form, I suggest that this is broken up to make it easier for the reader.

Was the evidence of heterogeneity in the MR just that rs159960 had a greater effect? If so, I think this can be motioned in the main text, and if this if this is the case, was that SNP the only one removed by MR-PRESSO? In that situation it seems that the MR-PRESSO result would be the optimal ones, and thus

the ones highlighted in the main text.

Supplementary figures 4 and 5 need larger, clearer labels.

When the authors say "We found concordant variation to be more predictive in females and positively correlated with other neuropsychiatric traits. Conversely, discordant variation was only significantly predictive in males and negatively correlated with most neuropsychiatric traits." do they have any suggestions as to why this should be? A lot of the discussion appears to be re-stating the results without much suggestion as to reasons for these results, or how this might be explored more specifically.

It would also be use to say if there are any hypostasized links or exist literature that might shed some light on the brain regions highlighted.

Reviewer #2 (Remarks to the Author):

The research presented in this paper addresses an important epidemiological and clinical topic: the association between loneliness/isolation and schizophrenia.

The authors used genome-wide summary data from large studies to test the hypothesis that polygenic scores for loneliness/isolation predict schizophrenia in an independent case-control study. The authors also parsed the genetic signal from summary statistics and calculated genetic correlations between phenotypes. Finally, they performed two-sample mendelian randomization to explore the causality of the association between loneliness/isolation and schizophrenia.

I have struggled to follow the manuscript, mostly because the way it is written is different from the academic English I am used to.

That may somehow hamper my input to the manuscript.

Methodology:

- Please describe and define the phenotypes and the characteristics of the samples in the training sets.

Were people with mental disorders excluded from the GWASs of loneliness/social isolation?

- I think there are larger schizophrenia GWASs that can be used (for example PMID: 29483656)

- I am not sure about the validity and meaning of the dissection of schizophrenia summary stats. Is there any previous study using this approach? Have authors considered Genomics SEM (PMID: 30962613)?

- For Mendelian randomization analyses, how many instrumental variables (genome wide significant SNPs) were considered?

- Was the MHC region included in the analyses?

Interpretation:

- Schizophrenia is genetically associated with a number of traits, often seen as not desirable. What does this study add to the previous literature (for example PMID: 27818178)?

- Social and occupational dysfunction is a diagnostic criterion for schizophrenia. As suggested in the

cited literature, loneliness is a complex construct. How do the definitions in the training GWAS compare to more sophisticated definitions and assessments?

Reviewer #3 (Remarks to the Author):

This paper presents a genetic investigation into the relationship between loneliness/ social isolation and schizophrenia. Dissecting their relationship is of high importance as loneliness/ social isolation is a potentially modifiable risk factor, relevant to the prevention or management of schizophrenia. I commend the authors for doing a thorough job of multiple testing correction, using a range of Mendelian Randomization methods and for using multiple tests to rule out horizontal pleiotropy. I have a couple of major points which need to be addressed and some minor points to improve the manuscript. Major:

It is essential to convert Nagelkerke's pseudo-R² to liability scale to facilitate comparison across the analyses conducted within this study and with other studies. Nagelkerke's pseudo-R² is biased by the proportion of cases in the testing dataset and prevalence of the disease in the population. Therefore, the analyses in males and females here cannot be compared using Nagelkerke's pseudo-R².

Transformation to the liability scale corrects this bias and can be performed using the procedures outlined by Lee et al., PMID: 22714935.

I have concerns about the power of the Mendelian Randomization analyses. I couldn't find any details on the P value used to select the instruments. As a rule of thumb, a robust MR analysis needs 10 independent genome-wide significant instruments. For the analysis of the effects of loneliness and isolation related traits on schizophrenia, the effect sizes are very inconsistent across the methods used (eg for LNL-ISO MTAG on SCZ the effect is in the opposite direction for MR Egger versus other methods which is quite concerning). For the effect of SCZ on loneliness/isolation related traits, the effect sizes are very consistent across methods (69 instruments used), suggesting this analysis is well-powered and the results are robust. My recommendations are: only test bidirectional relationships where a sufficient number of exposure instruments are available in both directions, use the best-powered method as the primary analysis and test for consistency of effect sizes across other methods. Does the MRBase package not eliminate outlier SNPs and retest and if not, can this be done after removing the SNPs identified as outliers via MR-PRESSO?

The direction of the SCZ_LNL_DISC annotation on each of these phenotypes is unclear. In interpreting the results of the genetic covariance between SCZ and other disorders for the SCZ_LNL_DISC annotation, could the authors clarify the direction of effect on SCZ? ie. Does this say that alleles that increase risk of SCZ and decrease LNL-ISO are positively correlated with BIP and negatively correlated with MDD? This is also relevant for the PRS SCZ_LNL_DISC – is the PRS increased or decreased in SCZ cases versus controls?

Minor:

On Figure 1 and 2 the y axes need proper labels – are these % R²s? For 1B the y axis is incorrect.

Fig 1B – including the results for the full SCZ PRS in the CIBERSAM sample would be helpful for comparison. I would also recommend getting the maximum number of SNPs in the PRS and in the h² calculations onto this Fig and into the results section (it only appears in the methods), because this is

highly relevant to the size of the R2s.

Fig 3 – I don't see the need to split disorders and traits into two panels. I think a single figure is appropriate here. It would help to make the colors more distinct from each other. You could use the same palette as Fig 1C for consistency.

Table 1 – What is the Heterogeneity Q Value referring to? Where are the results using the simple mode and weighted mode methods?

The CIBERSAM sample was genotyped as part of the Psychiatric Genomics Consortium. I trust the authors have ensured there is no overlap with the PGC SCZ GWAS summary statistics used to train the PRS, but please state this in the methods. Which reference panel was used to perform LD clumping for the PRS?

The abbreviations are a bit inconsistent eg LNL-ISO vs SCZ_noLNL.

I suggest using the terminology "Annotation-stratified genetic covariance" instead of "Partial genetic covariance" for consistency with the GNOVA paper and it's a more accurate description of the analysis. The word partial suggests that individual genomic loci are being examined.

I find the PRS for SCZ-LNL difficult to interpret. It's composed of a roughly equal number of CONC and DISC SNPs (slightly more CONC). Based on the results presented in Fig 2 I would expect either no difference between males and females in Fig 2B or higher PRS for SCZ_LNL in females, consistent with the higher number of CONC SNPs and the results in Fig 1C. I think these apparent inconsistencies could be due to the use of Nagelkerke's pseudo-R2, but nevertheless I still struggle with the interpretation of any findings using the PGS SCZ_LNL. Can the authors explain this?

It would be good to include the estimate of the overall genetic correlation between LNL-ISO and SCZ in the introduction.

Is the gamma from CAUSE equivalent (same interpretation and scale) to the beta from the other MR methods?

Reviewer #1 (Remarks to the Author):

This was an interesting paper, which performed a number of novel analyses, it was however lacking a clear thread to the argument, it was not clear how different parts of the paper related to each other, and some of the results were not fully discussed. Thus is it not always clear why the analysis is done, and what has been learnt from it. There was also a general tendency towards sentences that resembled lists and frequent acronyms (often with slightly ugly "_" in them). Together these made it hard to follow in places. A few specific comments are below:

We thank the reviewer for these comments. Per the reviewer's suggestions the manuscript has undergone extensive rewriting to increase clarity. We have also modified the acronyms to facilitate reading.

- **The results section needs a more sentence to give a high-level explanation rather than immediately pointing to a figure. Bearing in mind that the methods section is after the main text, some context for what is being done and why is needed.**

We thank the reviewer for this suggestion. We have modified the Results section to include more information on the analytical approach, as follows:

"We calculated polygenic scores for loneliness and isolation (PGSLNL-ISO) using the summary statistics from the combined multi-trait GWAS (MTAG) in the UK Biobank (UKBB) study. Figure 2A shows the percentage of variance in schizophrenia risk explained by LNL-ISO (PGSLNL-ISO) in the independent CIBERSAM case-control sample (NSCZ = 1927; NHC = 1561)."

In addition, we have included an additional figure (**Figure 1**) displaying the study workflow to facilitate the visualization of the whole analytic pipeline of the study.

- **What was the rationale for doing permutation tests for the PGS to schizophrenia? It seems unnecessary.**

We thank the reviewer for this comment. As the reviewer appropriately points out, these tests did not provide added value to our analyses and we have therefore decided to remove the permutation tests from the manuscript and supplementary info.

- **If you are going to talk about the relative size of an R2 (as done in line 164) it would make sense to report the CIs.**

We thank the reviewer for this insightful suggestion. Per the reviewer's suggestion we have now included 95% confidence intervals after bootstrap resampling for each R2 calculated through the manuscript. We describe this in the Methods section as follows:

“CIs for the increase in R^2 were estimated with bootstrap resampling ($N = 5000$ permutations), applying the Normal Interval method, after checking the normality of the bootstrap distribution.”

- **Figure 1 - I think it would be easier to have figures that didn't use quite so many acronyms, I think it would be better to spell out the factors. I don't like the formatting in part C of having a legend in place of axis labels.**

We thank the reviewer for this suggestion. We have now renamed the genomic partitions throughout the manuscript (including the Figures) in an attempt to improve readability, as follows:

subset of variants from Schizophrenia GWAS summary statistics not associated with LNL-ISO ($P_{\text{LNL-ISO}} > 0.05$) = SCZ[noLNL]

subset of variants from Schizophrenia GWAS summary statistics associated with LNL-ISO ($P_{\text{LNL-ISO}} < 0.05$) = SCZ[LNL]

subset of variants from Schizophrenia GWAS summary statistics associated with LNL-ISO ($P_{\text{LNL-ISO}} < 0.05$); with a concordant sign of the allele effect in both schizophrenia and LNL-ISO = SCZ[CONC]

subset of variants from Schizophrenia GWAS summary statistics associated with LNL-ISO ($P_{\text{LNL-ISO}} < 0.05$); with a discordant sign of the allele effect in both schizophrenia and LNL-ISO = SCZ[DISC]

Per the reviewer's suggestion we have also modified the Figure (now **Figure 2C**) to include axis labels and removed the legend accordingly, as follows:

- **I think the idea of splitting the data into sets based on the associations with loneliness needs to be explored a little more in the main text of the results - currently it is described in the legend to Figure 1, which doesn't seem like the optimal place.**

We thank the reviewer for this suggestion. We have now included a workflow chart (**Figure 1**) to clarify the genomic dissection pipeline.

We have also included a more detailed description in the Methods and Results sections of the main manuscript, as follows:

Results:

“Polygenic dissection of schizophrenia by its relationship with LNL-ISO”:

“Since $PGS_{LNL-ISO}$ encompassing variants with $PGS_{LNL-ISO} > 0.05$ did not contribute to schizophrenia risk (R^2 (95% CI) = 0.052% (-0.09,0.10) at $P_{threshold} > 0.05$, $p = 0.57$); supplementary data 1),, schizophrenia summary statistics were divided according to their role in LNL-ISO GWAS (Supplementary Methods 4): those variants not associated with LNL-ISO (SCZ[noLNL]; $P_{LNL-ISO} > 0.05$), those associated with LNL-ISO, with a concordant sign of the allele effect in both schizophrenia and LNL-ISO (SCZ[CONC]; $P_{LNL-ISO} < 0.05$; $Beta_{SCZ} > 0$ & $Beta_{LNL-ISO} > 0$ / $Beta_{SCZ} < 0$ & $Beta_{LNL-ISO} < 0$) and those associated with LNL-ISO but with a discordant direction of the effect relative to schizophrenia (SCZ[DISC]; $P_{LNL-ISO} < 0.05$; $Beta_{SCZ} > 0$ & $Beta_{LNL-ISO} < 0$ / $Beta_{SCZ} < 0$ & $Beta_{LNL-ISO} > 0$).”

Methods:

“First, we selected variants that were included in both schizophrenia and LNL-ISO summary data. Second, we divided schizophrenia summary statistics from the set of overlapping variants ($N_{SNPs} = 5,658,282$) into two different subsets of variants, according to their effects in the LNL-ISO (Figure 1): those variants not associated with LNL-ISO (SCZ[noLNL]; $P_{LNL-ISO} > 0.05$; $N_{SNPs} = 5,172,017$) and those variants associated with LNL-ISO (SCZ[LNL]; $P_{LNL-ISO} < 0.05$; $N_{SNPs} = 486,265$). We selected this cutoff based on the fact that LNL-ISO-based PGS ($PGS_{LNL-ISO}$) prediction on schizophrenia in the case-control target sample from CIBERSAM at $P_{LNL-ISO} > 0.05$ was not significant (R^2 (CI95%) = 0.052% (-0,09,0,19) at $P_{threshold} > 0.05$, $P = 0.569$; Supplementary Data 1 C). Third, based on the concordance or discordance of the effects of the same effect allele, we again divided SCZ[LNL] into those variants with concordant (SCZ[CONC]; $N_{SNPs} = 269,361$) or discordant (SCZ[DISC]; $N_{SNPs} = 216,904$) effects in schizophrenia and LNL-ISO (SCZ[CONC]: $Beta_{SCZ} > 0$ & $Beta_{LNL-ISO} > 0$ OR $Beta_{SCZ} < 0$ & $Beta_{LNL-ISO} < 0$; SCZ[DISC]: $Beta_{SCZ} > 0$ & $Beta_{LNL-ISO} < 0$ OR $Beta_{SCZ} < 0$ & $Beta_{LNL-ISO} > 0$). In each of the final datasets, we removed correlated SNPs due to linkage disequilibrium (LD) using PLINK 1.9 clumping algorithm ($r^2 > 0.1$; window size = 500 kb). See Supplementary Methods for further details.”

In addition, we have created a new Supplementary Methods subsection entitled “Supplementary Methods 4. Genomic dissection of schizophrenia based on LNL-ISO and stratified-polygenic score predictions”, which shows a detailed description of these methods.

- **I wonder if the comparative measures of heritability in lines 197-204 is an example of winners' curse? I.e. that the SNPs that occur in the loneliness data are simply more powered. What is the estimated heritability of the top 3.8% of the SNPs in SCZ-noLNL?**

We thank the reviewer for this insightful comment, which may contribute to ruling out possible artifacts and strengthening our findings. When we compare the top 3.8% of the SCZ[noLNL] partition with the complete SCZ[CONC] partition, we find higher enrichment estimates for the former. This is not surprising, as the top 3.8% of the SNPs within SCZ[noLNL] harbor variants more strongly associated with SCZ than those included SCZ[CONC] as we are comparing a subset of the most strongly associated variants with a complete partition.

Per the reviewer's comment we have now tested whether our finding of a heritability enrichment in SCZ[CONC] could be artifactually inflated as a consequence of the selection of SNPs that are significantly associated with loneliness ($P < 0.05$). If the heritability enrichment was caused by the selection of SNPs associated with loneliness, that inflating effect would be also observed in SCZ[DISC], which also harbors only SNPs associated with loneliness. Since SCZ[DISC] includes only 3.0% of SNPs in comparison to the 3.8% of SCZ[CONC], we tested this hypothesis by comparing the enrichment value of the SCZ[DISC] partition against the enrichments of 3.0% of the SNPs in the SCZ[CONC] partition, after 1,000 permutations with replacement.

All generated subsets of variants within SCZ[CONC] encompassing 3% of the SNPs lead to a higher heritability enrichment than that of SCZ[DISC], which also includes 3% of the SNPs. We believe that these results demonstrate that concordant SNPs are likely to be more enriched in predisposing variation to schizophrenia and that this finding is not merely due to their belonging to the variation associated with LNL-ISO ($P_{LNL-ISO} < 0.05$), since both SCZ[CONC] and SCZ[DISC] belong to that variation, but we do not find such enrichment for SCZ[DISC].

We have now incorporated a new Figure to illustrate this comparison (see Supplementary Figure 1). Moreover, we have also included a detailed description of this procedure in the Results section of the main manuscript and the Supplementary Methods section, as follows:

Results:

Polygenic dissection of schizophrenia by its relationship with LNL-ISO in results section

"Enrichment comparison of the same number of variants from SCZ[CONC] and SCZ[DISC] reflects a clearly superior enrichment of concordant variants (see Supplementary Figure 1 for further details)"

In Supplementary methods 4 (page 8):

"We are aware of the risk of inflated heritability enrichment values of annotations encompassing variants significantly associated with LNL-ISO due to an increase in power. We made an additional analysis to rule out this possibility by comparing the enrichment of the same number of variants from SCZ[CONC] and SCZ[DISC]. Since SCZ[DISC] includes only 3.0% of SNPs in comparison to the 3.8% of SCZ[CONC], the enrichment value of

SCZ[DISC] was compared against the enrichments of 3.0% of SNPs from SCZ[CONC] partition, after 1000 permutations with replacement from the whole set of SNPs in SCZ[CONC]. The distribution of the enrichments of all the subsets of 3% of SNPs within SCZ[CONC] was compared to the real enrichment of SCZ[DISC]. The P value was calculated by dividing the number of permuted SCZ[CONC] sets that explained equal or lower variance than the true SCZ[DISC] set plus one by the total number of permutations plus one. The results of this comparison are shown in Supplementary Figure 1."

- **Figure 2 it would be useful to see the means and CIs for this in a supplementary table.**

We thank the reviewer for this suggestion. We have now included the average R2 values and their corresponding 95% CIs after bootstrap resampling for the sex-stratified Spanish cohort in a supplementary Table (see Supplementary Data 4).

- **Figure 3 More detail on what "Cross-disorder phenotype" is.**

We appreciate the reviewer for this suggestion. Given the singularity of this GWAS meta-analysis, we have specified the details of this construct in the Methods section, as follows:

"Annotation-stratified genetic covariance "

"(...) cross-disorder phenotype (CROSS-DIS) - a meta-analysis across eight mental disorders with a total sample of 232,964 cases and 494,162 controls -"

We have also included additional details in Supplementary Methods 6:

"(...) a meta-analysis conducted by a specific PGC workgroup, with a total sample of 232,964 cases and 494,162 controls, among eight mental disorders (anorexia nervosa, attention-deficit/hyperactivity disorder, autism spectrum disorder, bipolar disorder, major depression, obsessive-compulsive disorder, schizophrenia, and Tourette syndrome) to analyse the pleiotropic contribution of different loci"

- **Lines 261-264, this is very list-y sentence, and it hard to follow in it's current form, I suggest that this is broken up to make it easier for the reader. Was the evidence of heterogeneity in the MR just that rs159960 had a greater effect? If so, I think this can be motioned in the main text, and if this if this is the case, was that SNP the only one removed by MR-PRESSO? In that situation it seems that the MR-PRESSO result would be the optimal ones, and thus the ones highlighted in the main text.**

We would like to thank the reviewer for these suggestions. Per the reviewer's suggestion we have now redrafted the paragraph and Table 1 reporting the results from the MR analysis.

As the reviewer appropriately points out, heterogeneity in the Loneliness UKBB analysis was influenced by the greater effect of rs159960. In the analyses for LNL-ISO, there was a similar scenario with 3 outliers due to greater effects, especially rs44655966. We

agree with the reviewer that in this context, the MR-PRESSO results should be highlighted in the Results section of the main text, and have done so, as follows:

“Mendelian randomization

We found evidence for a strong bi-directional causal effect of LNL-ISO on schizophrenia (WM - β (standard error (se)) = 1.37 (0.40), $p = 6.14 \times 10^{-4}$) with a consistent direction of the effects across methods except in the case of MR-Egger. The primary analysis method is the IVW method because it is the most efficient analysis with valid instrumental variables⁴⁵. However, its results may be biased if heterogeneity or the presence of pleiotropy⁴⁶ undermine the method’s assumptions. Although the MR-Egger intercept analysis did not detect horizontal pleiotropy ($p = 0.36$), we observed evidence of heterogeneity (IVW Q-p value = 2.94×10^{-6}) (Table 1, Supplementary Figure 4), which justifies the preference for the WM method. The presence of heterogeneity provided the most suitable explanation for the difference in the direction of Egger’s effect due to the sensitivity of this method to the presence of outliers and heterogeneity, which lead to poor causal estimates under that scenario. (see Table 1 and Supplementary Methods 7).

Additional robust methods that eliminate outliers that may be influencing the outcome due to pleiotropy (MR-PRESSO⁴⁷) or account for both correlated and uncorrelated pleiotropy (CAUSE⁴⁸) showed comparable results to those using the WM method, with even larger effect sizes using MR-PRESSO (MR-PRESSO outlier-correction β (Sd) = 1.45(0.30), $p = 0.001$; CAUSE - γ (CI95%) = 0.61 (0.34, 0.89), $p = 0.003$) (see Table 1).

We also found a causal effect of schizophrenia liability on LNL-ISO (WM - β (se) = 0.015(0.005), $p = 0.008$; CAUSE - γ (CI95%) = 0.01 (0.01, 0.01), $p = 0.003$), with evidence of heterogeneity (IVW Q-p value: 2.21×10^{-11}) but no indication, based on the MR-Egger intercept analysis, of horizontal pleiotropy ($p = 0.48$).

We found comparable evidence for bidirectional causality between Loneliness UKBB and schizophrenia to that found for LNL-ISO (Table 1, Supplementary Figure 4 and 5). We also found a unidirectional negative causal effect of ability to confide on schizophrenia (WM - β (se) = -0.6 (0.19), $p = 0.002$), and a unidirectional negative causal effect of schizophrenia on number of people in household (WM - β (se) = -0.011 (0.003), $p = 5.86 \times 10^{-3}$). We found no evidence of causality between the number of family/friends visits and schizophrenia. “

Similarly, we have modified the legend of supplementary figure 4:

“Note that a single variable (rs159960) in Loneliness UKBB on schizophrenia causes a greater discordant effect than the rest of the instruments in the leave-one-out test. Therefore, in this scenario, MRPRESSO produced a significant result after having eliminated outliers (See Table 1)”

In addition, we have included all the secondary and sensitivity analyses in a Supplementary Table (see Supplementary Data 7).

- **Supplementary figures 4 and 5 need larger, clearer labels.**

We thank the reviewer for these suggestions. We have now improved the quality and resolution of both figures (now Supplementary Figures 3 and 4).

- **When the authors say "We found concordant variation to be more predictive in females and positively correlated with other neuropsychiatric traits. Conversely, discordant variation was only significantly predictive in males and negatively correlated with most neuropsychiatric traits." do they have any suggestions as to why this should be? A lot of the discussion appears to be re-stating the results without much suggestion as to reasons for these results, or how this might be explored more specifically.**

We thank the reviewer for this comment. Per the reviewer's suggestion we have now expanded the paragraph discussing sex-related findings, as follows:

"Despite reported sex differences in the epidemiology and clinical manifestations of psychotic disorders^{36,38,57}, previous studies had not found an effect of sex on genetic associations⁵⁸. By analyzing the genomic overlap between schizophrenia and LNL-ISO, we did observe a differential effect of sex on polygenic predictions. Concordant overlapping variants in SCZ and LNL-ISO accounted for a significantly greater amount of variance in schizophrenia risk in females than in males, while the opposite pattern was observed in the rest of LNL-ISO based annotations. These results are in line with recent studies suggesting a potentially higher impact of loneliness and objective social isolation on psychiatric outcomes in females than in males^{26,41}. This may be due to a more negative perception of social deprivation in females related to their role in modern society⁵⁹ and a greater protective effect of an enriched social network in males⁶⁰. Moreover, among patients with schizophrenia, loneliness has been described to be more prevalent in females than males⁶¹. Our results suggest the existence of a social-related environment differentially affecting males and females that could be, at least in part, responsible for the different sex-stratified PGS predictions. Further studies should evaluate the impact of gender-based subjective social perception in epidemiological models. "

- **It would also be use to say if there are any hypostasized links or exist literature that might shed some light on the brain regions highlighted.**

We appreciate the reviewer's observation. We have now rephrased the Discussion section to discuss the involvement of the brain region with the higher LDSR enrichment

in the concordant subset of variants (i.e. the hippocampus) in social behavior and dysfunction, as follows:

“Researchers describe polygenic score predictions and LD-score-based partition heritability estimates as powerful methods for evaluating the effects of genetic predisposing variation within specific subsets of variants^{49–51}. With 3.8% of SNPs explaining an estimated 13.1% of SNP-based heritability, concordant overlapping variation between both phenotypes exhibits more than a three-fold increase in heritability enrichment compared to variants not predisposing to LNL-ISO, and a much higher enrichment than most of the genome-wide annotations previously evaluated in schizophrenia⁵². LDSC-SEG analyses pointed to a significant enrichment at the uncorrected level of concordant overlapping variation in the hippocampus, a brain region involved in social behavior^{53,54} and cognitive flexibility cognition⁵⁵, which may be especially sensitive to brain inflammation caused by loneliness and isolation^{10,54}. In this respect, recent work has described loneliness affecting white matter integrity of the hippocampus⁵⁶.”

Reviewer #2 (Remarks to the Author):

The research presented in this paper addresses an important epidemiological and clinical topic: the association between loneliness/isolation and schizophrenia.

The authors used genome-wide summary data from large studies to test the hypothesis that polygenic scores for loneliness/isolation predict schizophrenia in an independent case-control study. The authors also parsed the genetic signal from summary statistics and calculated genetic correlations between phenotypes. Finally, they performed two-sample mendelian randomization to explore the causality of the association between loneliness/isolation and schizophrenia.

I have struggled to follow the manuscript, mostly because the way it is written is different from the academic English I am used to. That may somehow hamper my input to the manuscript.

We thank the reviewer for their comments. The manuscript has undergone extensive revision and rewriting to address all the reviewers' comments and suggestions. We hope that this process has clarified the language.

Methodology:

- **Please describe and define the phenotypes and the characteristics of the samples in the training sets. Were people with mental disorders excluded from the GWASs of loneliness/social isolation?**

We thank the reviewer for this suggestion. Per the reviewer's suggestion, we have now added a Supplementary table including the descriptions of the training sets used (See Supplementary Data 6).

For the purposes of our study, we used the MTAG GWAS summary statistics from the study by Day et al., which did not exclude participants with mental disorders. In that study, the authors performed a sensitivity analysis excluding participants with self-reported depression, with no appreciable change in test statistic across any of the identified 15 *loci*, thus supporting that these *loci* do not influence loneliness through depression.

- **I think there are larger schizophrenia GWASs that can be used (for example PMID: 29483656)**

We would like to thank the reviewer for this comment. We decided to use SCZ PGC2 summary data, as we have worked with this data in previous studies (PMID: 32732888). PGC2 summary data have been also recently used in some papers of reference (for instance PMID: 31059627 or PMID: 33169016; PMID:30392411). We understand that these summary statistics have a solid validity for being used as discovery samples in polygenic score predictions or other related applications.

The sample from the PGC2 + clozuke study (Pardiñas et al., 2018) adds approximately 5000 new cases and 18000 controls to PGC2. The increase in sample size has proven successful for new GWAS *loci* discovery, although few improvements have been observed in terms of variance explanation or polygenic score prediction. This can be clearly observed in the new SCZ-PGC3 GWAS (currently in preprint; <https://www.medrxiv.org/content/10.1101/2020.09.12.20192922v1>). With an increase in SCZ cases of more than 20,000, it has not led to substantial changes in terms of variance explained.

Considering the nature of our work and the focus on polygenic prediction and genomic dissection, we believe that the choice of PGC2 instead of PGC2+clozuke should not have significantly affected our results. To test this, we conducted supplementary PGS_{SCZ} predictions using PGC2+clozuke and compared the results with our previous results using PGC2 summary data. We found that the PGC2+clozuke data did not substantially improve the PGS prediction (see Supplementary Table 2F).

In addition, we conducted supplementary partitioned SNP heritability analyses using the main annotations based on the relationship between SCZ and LNL-ISO ((SCZ[noLNL], SCZ[CONC] and SCZ[DISC]) using PGC2+clozuke summary statistics. We found similar

results for the abovementioned annotations regardless of the summary statistics used (PGC2 only / PGC2 + Clozuk; see Supplementary Table 3A).

We have now modified the Methods section and Supplementary info to include these analyses, as follows:

Methods

“Another recent schizophrenia GWAS 81, including approximately 5000 new cases and 18000 controls to PGC2, was also used to rule out changes in risk predictions or heritability estimates compared to PGC-SCZ2 GWAS.”

Supplementary info

Supplementary Methods 3. Polygenic score predictions

“PGC-SZ21 GWAS summary statistics have demonstrated a solid validity for being used as discovery samples in polygenic score predictions^{5,6} Anyway, to compare predictions with another recent schizophrenia GWAS 7 including approximately 5000 new cases and 18000 controls to PGC2, PGSSCZ were again calculated using SNPs from this later GWAS also present in LNL-ISO GWAS. No great improvements were observed in terms of explained variance.”

Supplementary Methods 5. LD-score regression (LDSC) and partitioning SNP heritability

“We also used another recent SCZ GWAS summary statistics⁷ to check that no differences in the results were observed (Supplementary table 3A).”

- **I am not sure about the validity and meaning of the dissection of schizophrenia summary stats. Is there any previous study using this approach? Have authors considered Genomics SEM (PMID: 30962613)?**

We thank the reviewer for this comment. We are aware of the many methodological available options to study the polygenic overlap of complex disorders and the genomic dissection of a complex trait based on its relationship to another trait.

For the purposes of this work we decided to dissect common variants across the whole genome using a simple SNP subsetting approach based on the association with the trait under study: LNL-ISO. We based our study on several pieces of evidence of a genetic overlap between schizophrenia and LNL-ISO. Our subsetting approach is similar to Lam et al's (2019; PMID: 31374203) and other authors' approaches, who assessed the pleiotropic effects of variation predisposing to one trait on others by conducting simple

selections of SNPs based on p-values. In our case, we selected the variants based on their association with LNL-ISO using a p-threshold of 0.05 and ensured that variants not associated with LNL-ISO ($PGS_{LNL-ISO}$ with variants $P_{LNL-ISO} > 0.05$) did not explain schizophrenia risk in a prediction of $PGS_{LNL-ISO}$ on schizophrenia. This allowed us to create partitions which contained variants associated with LNL-ISO (SCZ[LNL], SCZ[CONC] and (SCZ[DISC]) and which were not (SCZ[noLNL]). We then used these SNP sets to analyze the genomic footprint of LNL-ISO on SCZ using other well validated methods such as LDSC or GNOVA.

Genomic SEM is a great analytic tool to perform a polygenic dissection with many subsequent analyses. Although the inclusion of this alternative method would go beyond the scope of this work, we have now mentioned in the Discussion section that alternative methods such as GSEM may help to confirm and expand our results in future studies, as follows:

“Moreover, other methods for genomic dissection such as Genomic SEM⁷⁸ could be used in future studies to strengthen the results presented here.”

- **For Mendelian randomization analyses, how many instrumental variables (genome wide significant SNPs) were considered?**

We thank the reviewer for allowing us to clarify this. Per the reviewer’s comment we have now included the number of instruments for each analysis in a separate column in Table 1 and included additional information on the criteria for selecting the instruments in the Methods section, as follows:

“We selected genome-wide significant SNPs at $p < 5 \times 10^{-8}$ except in the case of “number of people in household” due to insufficient number of instrumental variables (IV) at this threshold. We used a $p < 5 \times 10^{-6}$ instead.”

- **Was the MHC region included in the analyses?**

We thank the reviewer for allowing us to clarify this. The MHC region was excluded in all analyses performed with Polygenic Risk Scores, LDSR, and GNOVA, following customary practices. For Mendelian Randomization analyses, which use SNPs as the instrumental variables, the exclusion of MHC is not among the guideline recommendations (see PMID: 32760811). To avoid confusion, we added this information to the methods section and supplementary info as follows:

Methods

“Genetic variants within Major Histocompatibility Complex (MHC) were removed”

Supplementary info

“Due to the extremely complex LD pattern, genetic variants within Major Histocompatibility Complex (MHC) were removed (from 26Mb to 33Mb of chromosome 6).”

Interpretation:

- **Schizophrenia is genetically associated with a number of traits, often seen as not desirable. What does this study add to the previous literature (for example PMID: 27818178)?**

There has been a surge in the number of recent publications on behavioural genetics in recent years, as Abdellaoui rightly points out in his latest review (PMID: 33986517). However, the field is still in its very early stages. Multiple GWAS on SES, Educational attainment or income have appeared with increasingly larger samples thanks to well characterized cohorts such as UK biobank and finnGEN or the 23andMe collaborations. Given the current growth of the field, we believe that moving towards a better characterization of phenotypes based on clinical descriptions is an essential step to obtain more accurate results. Loneliness and social isolation, as pointed out by several systematic reviews in recent years (PMID: 29327166, 28369646), are two key elements both in the early stages and in the disease process of psychotic disorders. There is therefore evidence that isolation is an important contributor to the severity of the clinical course of schizophrenia, although the causal relationship is still unclear. A possible explanation could be the social defeat hypothesis (PMID: 33168111). Our paper provides initial results for disentangling the relationship between the subjective perception of loneliness and social disconnection in patients with schizophrenia from a genetic perspective as well as the causal relationship between both traits. Our study adds to previous evidence of the social determinants of mental health by the genetics of income, social deprivation, and assortative mating, among others (PMID: 27818178, PMID: 31844048, PMID: 31745073, PMID: 33414549)

We have now expanded the Discussion section to incorporate these aspects, as follows:

“Henceforth, previous studies assessing social determinants of poor mental health have evaluated the association of social disadvantage and their genetic determinants with the risk of psychosis. Our study adds to this previous evidence by incorporating a subjective perception to social dysfunction in psychosis from a genetic perspective. Further studies should explore the effect of subjective perception of loneliness and its association with the social defeat hypothesis with the risk of psychosis”

We believe that that our results open new avenues of research on the potentially leading role of subjective perception on causality of mental disorders, sex-specific effects of social determinants of mental health and the different pleiotropic patterns based on social connections and loneliness in mental disorders.

- **Social and occupational dysfunction is a diagnostic criterion for schizophrenia. As suggested in the cited literature, loneliness is a complex construct. How do the definitions in the training GWAS compare to more sophisticated definitions and assessments?**

We thank the reviewer for this comment. As the reviewer very appropriately points out, social dysfunction is found in all stages of schizophrenia. We decided to focus on loneliness and isolation because these are two complex constructs that are present in both clinical and non-clinical populations and they incorporate a subjective approach to social dysfunction in psychosis.

The UK Biobank used self-reported single-question questionnaires to assess loneliness and social isolation. Multiple research papers have validated self-reported binary loneliness questionnaires (PMID: 18504506, PMID: 660402), with strong convergent validity with validated scales such as the UCLA scale. A recently published neuroimaging paper also used the self-reported single-question of loneliness in UK-Biobank (PMID: 33319780). The original paper reporting the MTAG results for LNL-ISO (Day et. al. 2018) also tested the validity of the construct in an independent sample from the Health and Retirement Study (HRS) where they used a 3-item questionnaire: This questionnaire had been shown to be highly correlated with the UCLA-loneliness scale.

Nevertheless, we are aware of the fact that brief self-reported measures have limitations for capturing complex phenotypes such as loneliness and social isolation and have therefore acknowledged this as a significant limitation of the paper in the Discussion section of the manuscript, as follows:

“First, we used measures of loneliness and objective social isolation from the UKBB, which are based on single-question questionnaires and not on validated scales such as UCLA Loneliness. Although multiple research studies have previously validated binary self-reported loneliness questionnaires with strong convergent validity with UCLA Loneliness scale ”

Reviewer #3 (Remarks to the Author):

This paper presents a genetic investigation into the relationship between loneliness/ social isolation and schizophrenia. Dissecting their relationship is of high importance as loneliness/ social isolation is a potentially modifiable risk factor, relevant to the prevention or management of schizophrenia. I commend the authors for doing a

thorough job of multiple testing correction, using a range of Mendelian Randomization methods and for using multiple tests to rule out horizontal pleiotropy.

We thank the reviewer for these positive remarks.

I have a couple of major points which need to be addressed and some minor points to improve the manuscript.

Major:

- It is essential to convert Nagelkerke's pseudo-R² to liability scale to facilitate comparison across the analyses conducted within this study and with other studies. Nagelkerke's pseudo-R² is biased by the proportion of cases in the testing dataset and prevalence of the disease in the population. Therefore, the analyses in males and females here cannot be compared using Nagelkerke's pseudo-R². Transformation to the liability scale corrects this bias and can be performed using the procedures outlined by Lee et al., PMID: 22714935.

We especially thank the reviewer for this suggestion. We agree with the reviewer that a male/female bias in prevalence, which we had not previously considered in the male-female analyses across annotations, could have affected our previous results.

Per the reviewer's suggestion, we have converted all Nagelkerke-R² to the liability scale using the procedure described in Lee et al (PMID: 22714935) so as to make all PGS predictions comparable across the study and other similar studies. We have also calculated 95% confidence intervals for the liability R^2 estimates with bootstrap resampling so that PGS results are now reported as liability R^2 with 95% CI throughout the paper.

We have modified the Methods section to reflect these changes, as follows:

"Explained variance attributable to PGS was calculated as the increase in R^2 between a model with and without the PGS variable. Nagelkerke's pseudo-R² were converted to liability scale following the procedure proposed by Lee et al, assuming a prevalence of schizophrenia in the general population of 1%."

"CI for the increase in R^2 was estimated through bootstrap resampling ($N = 5000$ permutations), applying the Normal Interval method, after checking the normality of the bootstrap distribution."

We also converted R^2 estimates for the PGS_{SCZ} prediction models stratified by sex to the liability scale. For these analyses, prevalence data of schizophrenia in males and females were considered as 1%, since no clear differences in prevalence have been reported (PMID: 29762765.)However, as a sensitivity analysis, we also extracted prevalence estimates from a recent study performed in Spanish population that do report differences in prevalence of schizophrenia among males and females (0.0047 for females and 0.0079 for males; PMID:32248839). The results of the sex comparisons can be observed in Figure 3 and supplementary data 4. Also, supplementary Figure 3 showed similar results regardless of the alternative prevalence estimates considered. We have

modified the Methods and supplementary information sections to reflect this, as follows:

Methods

“we statistically compared the differences between the distribution of liability R2 in males and females across each genomic partition with two-sided t-tests. No sex differences in various prevalence measures have been reported^{84,85}. Therefore, prevalence of 1% has been considered for both sexes. As a sensitivity analysis, recent estimates reporting differences between sexes in Spanish population⁸⁶ (prevalence in males = 0.0079 and females = 0.0045) have been also considered and the analyses were repeated showing no differences in the sex comparisons (Supplementary Figure 3).”

Supplementary info

“ (...)So, we took a permutation-based approach: we performed bootstrap resampling (5,000 permutations) of 500 schizophrenia and 500 HC subjects across men and women separately within the SCZ_CIBERSAM cohort, and calculated PGS and explained variance predictions by logistic regression and R2 estimations on the liability scale. The P-threshold with the higher prediction in each sex was selected for each partition. No sex differences in various prevalence measures have been reported^{8,9}. Therefore, prevalence of 1% has been considered for both sexes¹⁰. As a sensitivity analysis, recent estimates reporting differences between sexes in Spanish population¹¹ (prevalence in males = 0.0079 and females = 0.0045) have been also considered and the analyses were repeated showing no differences in the sex comparisons (Supplementary Figure 3). The process was repeated for each of the SNP subsets: SCZ[noLNL], SCZ[LNL], SCZ[CONC] and SCZ[DISC]. Variance explained in women and men was statistically compared by student t-test.”

- **I have concerns about the power of the Mendelian Randomization analyses. I couldn't find any details on the P value used to select the instruments. As a rule of thumb, a robust MR analysis needs 10 independent genome-wide significant instruments. For the analysis of the effects of loneliness and isolation related traits on schizophrenia, the effect sizes are very inconsistent across the methods used (eg for LNL-ISO MTAG on SCZ the effect is in the opposite direction for MR Egger versus other methods which is quite concerning). For the effect of SCZ on loneliness/isolation related traits, the effect sizes are very consistent across methods (69 instruments used), suggesting this analysis is well-powered and the results are robust. My recommendations are: only**

test bidirectional relationships where a sufficient number of exposure instruments are available in both directions, use the best-powered method as the primary analysis and test for consistency of effect sizes across other methods. Does the MRBase package not eliminate outlier SNPs and retest and if not, can this be done after removing the SNPs identified as outliers via MR-PRESSO?

We would like to thank the reviewer for these comments and suggestions.

Per the reviewer's suggestions, we have used the default more stringent threshold of $P < 5 \times 10^{-8}$ for all traits that have at least 10 instrumental variables (IV) and re-analyzed the "Number of people in household" trait with a more permissive threshold to include more IVs with comparable results to those found with more stringent threshold (3 IVs). We have now added included this information and additional details of the MR analyses in the Methods section and the Table 1 legend, as follows:

Methods:

Two sample Mendelian Randomization

"We selected genome-wide significant SNPs at $p < 5 \times 10^{-8}$ except for the trait "number of people in household" due to insufficient number of instrumental variables (IV) at this threshold. We used a $p < 5 \times 10^{-6}$ instead. We applied a default LD-Clumping R^2 threshold of 0.001 and a window of 10000kb."

Table 1:

"By default we selected genome-wide significant SNPs as Instrumental Variables at $p < 5 \times 10^{-8}$ for all the traits except for "Number of people in household" due to a lower number of significant SNPs at that threshold. For this trait, we used a threshold of $p < 5 \times 10^{-6}$ instead."

The different sign of the MR-Egger effect sizes in the LNL-ISO analyses relative to other methods was one of our main concerns, which led us to include novel additional methods such as CAUSE to increase certainty of the direction of the effects. Due to its characteristics MR-Egger is the method most affected by the presence of heterogeneity and outliers, as it allows up to 100% of the IVs to be invalid and is therefore used predominantly as a sensitivity analysis (PMID: 28527048). Considering that the Egger intercept was not significant in our analyses, the consistency of the results of other sensitivity methods such as MRPRESSO and WM and the abovementioned limitations, MR-Egger would not be a method of choice (PMID: 32760811) and that results from other methods are more reliable in this scenario. We have now modified the Results section to reflect this, as follows:

"We found evidence for a strong bi-directional causal effect of LNL-ISO on schizophrenia (WM - β (standard error (se)) = 1.37 (0.40), $p = 6.14 \times 10^{-4}$) with a consistent direction of the effects across methods except in the case of MR-Egger. IVW method is the primary analysis method, because it is the most efficient analysis method with valid instrumental variables but results may be biased if not all assumptions were met due to heterogeneity or the presence of pleiotropy. Although the MR-Egger intercept analysis did not detect horizontal pleiotropy ($p = 0.36$), we observed evidence of heterogeneity (IVW Q-p value

= $2,94 \times 10^{-6}$), which justifies the preference for the WM method. The presence of heterogeneity provided the most suitable explanation for the difference in the direction of Egger's effect due to the sensitivity of this method to the presence of outliers and heterogeneity, which lead to poor causal estimates under that scenario. (see Table 1 and Supplementary Methods 7). Additional robust methods that eliminate outliers that may be influencing the outcome due to pleiotropy (MR-PRESSO) or account for both correlated and uncorrelated pleiotropy (CAUSE) showed comparable results to those of the WM method, with even larger effect sizes using MR-PRESSO (MR-PRESSO outlier-correction β (Sd) = 1.45(0.30), $p = 0.001$; CAUSE - γ (CI95%) = 0.61 (0.34, 0.89), $p = 0.003$) (see Table 1). "

MR-PRESSO Beta-Effect sizes in Table 1 are calculated after removing outliers. We have now added this information to Table 1 legend, as follows:

"The column "Outliers" reports the number of pleiotropic variants removed from the MR estimate. MR-PRESSO β -Effects were estimated after removing the outliers."

- **The direction of the SCZ_LNL_DISC annotation on each of these phenotypes is unclear. In interpreting the results of the genetic covariance between SCZ and other disorders for the SCZ_LNL_DISC annotation, could the authors clarify the direction of effect on SCZ? ie. Does this say that alleles that increase risk of SCZ and decrease LNL-ISO are positively correlated with BIP and negatively correlated with MDD? This is also relevant for the PRS SCZ_LNL_DISC – is the PRS increased or decreased in SCZ cases versus controls?**

We thank the reviewer for these insightful comments.

In order to provide readers with a direct interpretation of the direction of effects we have tried to clarify the methods for defining the SCZ[CONC] and SCZ[DISC] partitions in a new Figure (Figure 1) and have now included a quantile plot of PGS_{SCZ} in Figure 2C. As it can be seen in Figure 2C, every partition has the same direction of effects on schizophrenia risk, with higher PGS_{SCZ} observed in cases relative to HC. This is also the case for SCZ[DISC], so that overlapping variants with discordant effects in SCZ and LNL-ISO (i.e. that increase the risk for schizophrenia and decrease LNL-ISO) are also associated with increased schizophrenia risk.

In the case of covariance analyses, results for the SCZ[DISC] partition indicate that alleles that increase risk of SCZ and decrease LNL-ISO are positively correlated with BIP and negatively correlated with MDD risk.

We have tried to further clarify this in the Results section, as follows:

Results

"Annotation-stratified genetic covariance between SCZ and related phenotypes based on LNL-ISO:

The majority of the disorders (MDD, ANX, ADHD, ASD, CROSS-DIS, ALC-DEP) and personality traits (NEUR, SWB, DS, PSY_EXP) tested here showed genetic positive correlation within SCZ[CONC] and genetic negative correlation within SCZ[DISC] (Figure 4). However, BIP and OCD showed positive covariances within both genomic annotations. Therefore, alleles that increase risk for SCZ but decrease risk for LNL-ISO (SCZ[DISC]) are positively correlated with BIP or OCD, but negatively correlated with MDD, ASD, ADHD or ANX, providing one distinction between these two groups in their relation with LNL-ISO.”

We have also expanded the interpretation of these results in the Discussion section by including a reference from a study dissecting the genetic relationship between psychiatric disorders using GSEM (PMID: 32931100), with consistent results to ours in regards to stratified covariances, as follows:

“However, OCD and BIP are positively correlated with schizophrenia regardless of LNL-ISO based annotations, thus suggesting that the association of these disorders with schizophrenia is independent from the genetic predisposition to LNL-ISO. These results are in line with recent findings suggesting that schizophrenia, BIP, and OCD could belong to the same psychopathology factor at the genomic level.”

Minor:

- **On Figure 1 and 2 the y axes need proper labels – are these % R2s? For 1B the y axis is incorrect.**

We thank the reviewer for this suggestion. Per the reviewer’s suggestion, we have now modified the Figures (now Figure 2 and 3) to include axis labels.

- **Fig 1B – including the results for the full SCZ PRS in the CIBERSAM sample would be helpful for comparison. I would also recommend getting the maximum number of SNPs in the PRS and in the h2 calculations onto this Fig and into the results section (it only appears in the methods), because this is highly relevant to the size of the R2s.**

We thank the reviewer for these suggestions. Per the reviewer’s suggestion, we have modified the Figure (now Figure 2) to incorporate the results of the full SCZ PRS (shown as SCZ[ALL]). We have also incorporated a histogram and included the maximum number of SNPs in each genomic partition in Figure 2B.

- **Fig 3 – I don’t see the need to split disorders and traits into two panels. I think a single figure is appropriate here. It would help to make the colors more distinct from each other. You could use the same palette as Fig 1C for consistency.**

We thank the reviewer for this suggestion. Per the reviewer’s suggestion, we have included all the disorders and traits in a single panel. We have also revised all the Figures

of the manuscript using the same color palette for the partitions to increase consistency and facilitate interpretation.

- **Table 1 – What is the Heterogeneity Q Value referring to? Where are the results using the simple mode and weighted mode methods?**

We thank the reviewer for allowing us to clarify this, as there was an error in the previous version of the manuscript. Table 1 now shows Q p-value from the heterogeneity tests (IVW and Egger Cochran 's Q statistic) that can be performed with the Two-sample MR package. We have now mentioned these analyses in the Methods section, as follows:

“Different sensitivity analyses such as heterogeneity tests (IVW and Egger Cochran 's Q statistic test), leave-one-out and pleiotropy tests were performed using functions of the same R package.”

And Table 1:

“Q-P Value, P value of IVW Cochran 's Q statistic.”

Per the reviewer's question of Simple mode and Weighted mode methods, we have included the results of all methods in the Supplementary Data 7.

Further details on the methods, statistics and results of the MR analyses can be found in the Supplementary Material (Supplementary Methods 7, and Supplementary Data 7).

- **The CIBERSAM sample was genotyped as part of the Psychiatric Genomics Consortium. I trust the authors have ensured there is no overlap with the PGC SCZ GWAS summary statistics used to train the PRS, but please state this in the methods. Which reference panel was used to perform LD clumping for the PRS?**

We thank the reviewer for allowing us to clarify this. As the reviewer appropriately points out, there is no overlap between the discovery sample from the Psychiatric Genomics Consortium - SCZ2 (Ripke et al., 2014) and the CIBERSAM sample, which was genotyped more recently for PGC3 (currently in pre-print). We have now mentioned this explicitly in the Methods section, as follows:

“There is no overlap between PGC-SCZ2 and SCZ_CIBERSAM samples”

LD-clumping was performed using European data from 1000 genomes. We have included this information in Supplementary Methods 3, as follows:

“LD reference from the European subset of the 1000 Genomes reference panel was used from clumping”

- **The abbreviations are a bit inconsistent eg LNL-ISO vs SCZ_noLNL.**

We thank the reviewer for this suggestion. We have revised the paper to increase consistency in the acronyms used, and have modified them to facilitate reading, as follows:

subset of variants from Schizophrenia GWAS summary statistics not associated with LNL-ISO ($P_{\text{LNL-ISO}} > 0.05$) = SCZ[noLNL]

subset of variants from Schizophrenia GWAS summary statistics associated with LNL-ISO ($P_{\text{LNL-ISO}} < 0.05$) = SCZ[LNL]

subset of variants from Schizophrenia GWAS summary statistics associated with LNL-ISO ($P_{\text{LNL-ISO}} < 0.05$); with a concordant sign of the allele effect in both schizophrenia and LNL-ISO = SCZ[CONC]

subset of variants from Schizophrenia GWAS summary statistics associated with LNL-ISO ($P_{\text{LNL-ISO}} < 0.05$); with a discordant sign of the allele effect in both schizophrenia and LNL-ISO = SCZ[DISC]

- **I suggest using the terminology “Annotation-stratified genetic covariance” instead of “Partial genetic covariance” for consistency with the GNOVA paper and it’s a more accurate description of the analysis. The word partial suggests that individual genomic loci are being examined.**

We thank the reviewer for the recommendation. We have now replaced “partial genetic covariance” with “Annotation-stratified genetic covariance” throughout the manuscript

- **I find the PRS for SCZ-LNL difficult to interpret. It’s composed of a roughly equal number of CONC and DISC SNPs (slightly more CONC). Based on the results presented in Fig 2 I would expect either no difference between males and females in Fig 2B or higher PRS for SCZ_LNL in females, consistent with the higher number of CONC SNPs and the results in Fig 1C. I think these apparent inconsistencies could be due to the use of Nagelkerke’s pseudo-R², but nevertheless I still struggle with the interpretation of any findings using the PGS SCZ_LNL. Can the authors explain this?**

We thank the reviewer for allowing us to clarify this. Per the reviewer’s recommendation, we now report the variance explained on the liability scale throughout the paper, now represented in **Figure 3**.

Overall, there is a greater variance explained by PGS analyses in males than in females (either SCZ[noLNL] or SCZ[LNL]). Only concordant overlapping variants in SCZ and LNL-ISO (SCZ[CONC]) accounted for a significantly greater amount of variance in schizophrenia risk in females, while the opposite pattern was observed in the rest of LNL-ISO based annotations.

The aspects of the overall greater variance explained by the male case-control subsample are beyond the scope of the present work. Future studies should explore

whether this difference could be attributed to the nature of the CIBERSAM sample, a likely male-biased SCZ phenotype or a higher load of environmental factors affecting women, among other factors. Therefore, we focused our interpretation on the particular results within the SCZ[CONC] partition, that reflect a different pattern of variance explained in each of the sexes relative to that of the remaining partitions. These results suggest a greater role of genetic predisposing variation to loneliness and social isolation in schizophrenia risk in females.

We have now modified the Discussion section to reflect this, as follows:

“ Despite reported sex differences in the epidemiology and clinical manifestations of psychotic disorders^{36,38,57}, previous studies had not found an effect of sex on genetic associations⁵⁸. By analyzing the genomic overlap between schizophrenia and LNL-ISO, we did observe a differential effect of sex on polygenic predictions. Concordant overlapping variants in SCZ and LNL-ISO accounted for a significantly greater amount of variance in schizophrenia risk in females than in males, while the opposite pattern was observed in the rest of LNL-ISO based annotations. These results are in line with recent studies suggesting a potentially higher impact of loneliness and objective social isolation on psychiatric outcomes in females than in males^{26,41}. This may be due to a more negative perception of social deprivation in females related to their role in modern society⁵⁹ and a greater protective effect of an enriched social network in males⁶⁰. Moreover, among patients with schizophrenia, loneliness has been described to be more prevalent in females than males⁶¹. Our results suggest the existence of a social-related environment differentially affecting males and females that could be, at least in part, responsible for the different sex-stratified PGS predictions. Further studies should evaluate the impact of gender-based subjective social perception in epidemiological models.”

- **It would be good to include the estimate of the overall genetic correlation between LNL-ISO and SCZ in the introduction.**

We thank the reviewer for this suggestion. Per the reviewer’s suggestion we have now included the estimate of the overall genetic correlation between loneliness and schizophrenia based on the study by Day et al (2018) in the Introduction section, as follows:

“This study also reported a significant genetic correlation of the combined phenotype (LNL-ISO) with schizophrenia ($r_g = 0.17$, $p = 3.47 \times 10^{-12}$), along the lines of a previous study reporting a significant association of perceived loneliness with schizophrenia, but not with bipolar disorder³⁰”

- **Is the gamma from CAUSE equivalent (same interpretation and scale) to the beta from the other MR methods?**

We thank the reviewer for allowing us to clarify this. The gamma from CAUSE is equivalent to the beta from MR methods in terms of effect direction but not in terms of scale. This is because the CAUSE method is based on the assumptions that the causal effect of M(exposure) on Y(outcome) leads to a correlation between $\beta_{M,j}$ and $\beta_{Y,j}$ for all variants with a nonzero effect on M, while a shared factor induces correlation for only a subset of M effect variants, as shown in the following equation:

$$\beta_{Y,j} = \underbrace{\gamma\beta_{M,j}}_{\text{causal effect}} + \underbrace{Z_j\eta\beta_{M,j}}_{\text{correlated pleiotropy}} + \underbrace{\theta_j}_{\text{uncorrelated pleiotropy}},$$

CAUSE produces one sharing model with γ fixed at 0, allowing for horizontal pleiotropic effects but no causal effect, and one causal model with γ as a free parameter. CAUSE then compares these models by using the expected log pointwise posterior density (ELPD), a Bayesian model comparison approach that estimates how well the posterior distributions of a particular model are expected to predict a new set of data. In CAUSE, γ is the result of the $\beta_{M,j}$ parameters in the causal model and the P value is based on the comparison between the two models(ELPD) (PMID: 32451458).

We have included additional information on CAUSE and the interpretation of the models in Supplementary Methods 7 and Table 1:

“ γ (CI95%) Posterior median and 95% credible intervals of true value of causal effect with CAUSE. P (CAUSE): p-value testing that causal model is better than sharing model using ELPD test (significance level $p < 0.05$)”

REVIEWERS' COMMENTS

Reviewer #1 (Remarks to the Author):

The authors have substantially improved the paper, and I thank them for the detailed responses to my comments. I do, however, have a small number of minor residual comments.

Re-reading the manuscript; I realised that it might confuse some readers that, for example, SCZ[LNL] refers to all the SNPs that matched up between the two GWAS files, and then filtered based on the loneliness results. I think some people might initially think that it refers to a set of SNPs that had some evidence of association with schizophrenia. For this reason it would be good to have a note on the relevant part of Figure 1 (similar to the "P LNL-ISO < 0.05") to make this clear.

Lines 209-218. This paragraph appears to be all one sentence. I think it needs to be broken up into smaller bits.

Line 307. There's a "," rather than a "." in the p-value quoted. This also happens with the p-values reported in Table 1.

Figure 2 D. There seems a printing error in the title where "2", "h" and "SNP" are all printed over each other (though this might only be in the review version).

Lines 427-430. I think it would be good to also mention the result shown in Figure 2 A, as that strengthens the point.

Supplementary Methods 4. The authors describe using a clumping metric. Usually when clumping we also use the p-values for an association with a phenotype, so that we can pick the "best" SNP. I'm guessing that the SNPs are clustered based on their association with schizophrenia, but it would be good to make this clear.

Reviewer #2 (Remarks to the Author):

some of my methodological concerns have been addressed.

Writing, however, still needs improvement, as it makes the thought process behind the research difficult to follow. The abstract especially requires attention. For example: in the abstract genetic variation is first singular then plural;

abstract is in present tense, text in past tense. Claims in the abstract are not supported by stats. Except if there is an explicit journal policy that requires to exclude stats from abstract, I strongly recommend their inclusion.

For example first result needs to be supported by stats showing the predictive power (ROC for example). If terminology suggesting "prediction" is used, this needs to be quantified in the text (again, with a ROC,

for example). Otherwise the R2 shows the proportion of variance explained by a variable of interest on another variable. Clarity on whether this research is about prediction versus explanation is fundamental and impacts the study design (see Shmueli 2010 famous paper on this). This point still does not come across.

In this context, the associations presented are definitely significant, but the effect sizes small.. this would need to be discussed (it suggests other variables, not considered here, may play a larger role both in prediction and in the causal/ explanatory pathways..

Please take the sentence on COVID 19 out, it is random, generic and not circumstantiated (it cites an opinion piece).

Reviewer #3 (Remarks to the Author):

The authors have done a good job of addressing my concerns.

I just have 2 more minor points.

Why is Nagelkerke's pseudo-R2 still reported in Fig 2A? This should be R2 on the liability scale, since the outcome variable is SCZ versus control status.

In Table 1, the column "Outliers" reports the number of pleiotropic variants removed from the MR estimate. I am not clear on which MR analysis this refers to.

REVIEWERS' COMMENTS

Reviewer #1 (Remarks to the Author):

The authors have substantially improved the paper, and I thank them for the detailed response to my comments. I do, however, have a small number of minor residual comments.

We thank the reviewer for these positive comments and appreciate the suggested revisions to improve the paper.

Re-reading the manuscript; I realised that it might confuse some readers that, for example, SCZ[LNL] refers to all the SNPs that matched up between the two GWAS files, and then filtered based on the loneliness results. I think some people might initially think that it refers to a set of SNPs that had some evidence of association with schizophrenia. For this reason it would be good to have a note on the relevant part of Figure 1 (similar to the "P LNL-ISO < 0.05") to make this clear.

We thank the reviewer for this suggestion. We have now modified Figure 1 to clarify that SNPs are included regardless of their association with SCZ by adding a note ($P_{SCZ} < 1$) to the previous notes describing SCZ[LNL] and SCZ[noLNL].

Lines 209-218. This paragraph appears to be all one sentence. I think it needs to be broken up into smaller bits.

We thank the reviewer for this suggestion. Per the reviewer's suggestion we have now redrafted the whole paragraph, as follows (lines 226-240) :

*"Since $PGS_{LNL-ISO}$ encompassing variants with $P_{LNL-ISO} > 0.05$ did not contribute to schizophrenia risk (R^2 (95% CI) = 0.052% (-0.09,0.10) at $P_{threshold} > 0.05$, $p = 0.57$); **Supplementary Data 1**), schizophrenia summary statistics were subsetted according to their role in LNL-ISO GWAS (**Supplementary Methods 4**). Firstly, those variants not associated with LNL-ISO (**SCZ[noLNL]**; $P_{LNL-ISO} > 0.05$) were extracted. Second, variants associated with LNL-ISO ($P_{LNL-ISO} < 0.05$) were divided into those with a concordant sign of the allele effect in both schizophrenia and LNL-ISO (**SCZ[CONC]**; $Beta_{SCZ} > 0$ & $Beta_{LNL-ISO} > 0$ / $Beta_{SCZ} < 0$ & $Beta_{LNL-ISO} < 0$) and those with a discordant direction of the effect relative to schizophrenia (**SCZ[DISC]**; $Beta_{SCZ} > 0$ & $Beta_{LNL-ISO} < 0$ / $Beta_{SCZ} < 0$ & $Beta_{LNL-ISO} > 0$; **Figure 1**).*

*We performed PGS_{scz} predictions on the same schizophrenia case-control sample for the three described subsets of SNPs: PGS_{scz} predictions from variants only contributing to SCZ ($PGS_{SCZ[noLNL]}$) and from those contributing to both phenotypes with concordant ($PGS_{SCZ[CONC]}$) and discordant ($PGS_{SCZ[DISC]}$) effects (see **Methods**)."*

Line 307. There's a "," rather than a "." in the p-value quoted. This also happens with the p-values reported in Table 1.

We thank the reviewer for this observation. We have corrected the p-values in both Table 1 and line 307 (now, line 329).

Figure 2 D. There seems to be a printing error in the title where "2", "h" and "SNP" are all printed over each other (though this might only be in the review version).

We thank the reviewer for this observation. We have now removed the printing error from Figure 2.

Lines 427-430. I think it would be good to also mention the result shown in Figure 2 A, as that strengthens the point.

We thank the reviewer for this comment. Per the reviewer's suggestion we have now expanded that paragraph in the Discussion to mention the results from Figure 2A, as follows:

"Causal inferences assessing the relationships between LNL-ISO constituents and schizophrenia also found a unidirectional negative causal effect of "ability to confide" on schizophrenia, in line with recent studies describing the association of lack of confidence and loneliness with psychosis, which may be mediated by negative schemata of others^{29,78}. Moreover, an unidirectional negative causal effect of schizophrenia liability on the "number of people living in the household" was found, thus suggesting a possible indirect causal effect of schizophrenia genetic liability on subsequent social disconnection in participants diagnosed with schizophrenia¹⁸. This relationship is also reinforced with the significant polygenic contribution of both phenotypes to schizophrenia risk (Figure 2A)"

Supplementary Methods 4. The authors describe using a clumping metric. Usually when clumping we also use the p-values for an association with a phenotype, so that we can pick the "best" SNP. I'm guessing that the SNPs are clustered based on their association with schizophrenia, but it would be good to make this clear.

We thank the reviewer for this observation. We have expanded the information in Supplementary Methods 4 to clarify that the clumping method is based on the selection of the LD-independent SNPs that are more associated with SCZ, as follows:

*"In each dataset, we removed correlated SNPs due to linkage disequilibrium (LD) using the PLINK 1.9 clumping algorithm based on the selection of the LD-independent SNP that were more associated with SCZ and we used independent variants within **SCZ[noLNL]** (N_clumpedSNPs = 169574), **SCZ[LNL]** (N_clumpedSNPs = 118,04), **SCZ[CONC]** (N_clumpedSNPs = 6468) and **SCZ[DISC]** (N_clumpedSNPs = 5,336) to calculate PGS on the SCZ_CIBERSAM case-control cohort (N_SCZ = 1,927; N_HC = 1,561)."*

Reviewer #2 (Remarks to the Author):

Some of my methodological concerns have been addressed.

We thank the reviewer for these positive comments, and appreciate the suggested revisions to improve the paper.

Writing, however, still needs improvement, as it makes the thought process behind the research difficult to follow. The abstract especially requires attention. For example: in the abstract genetic variation is first singular then plural; abstract is in present tense, text in past text. Claims in the abstract are not supported by stats. Except if there is an explicit journal policy that requires to exclude stats from abstract, I strongly recommend their inclusion.

We thank the reviewer for these suggestions. The Abstract and the main text have undergone extensive revision and rewriting to improve clarity in language and structure. We have also asked a native English speaker to proofread the main text and we have made several changes based on their suggestions. Moreover, we have revised the Results section extensively to account for the fact that the Methods section is placed at the end of the published article. We hope that these changes have helped to improve the readability of the manuscript.

We appreciate the reviewer's recommendation to include the stats in the Abstract. However, considering the high number of analyses described in the Abstract, we found it difficult to meet the word limit for this section if we included them. Therefore, we have modified the Abstract per the reviewer's and editorial suggestions, but we have preferred not to include any stats. We remain at the disposal of the editorial team, should they consider it advisable for us to include statistics in the Abstract.

For example, the first result needs to be supported by stats showing the predictive power (ROC for example). If terminology suggesting "prediction" is used, this needs to be quantified in the text (again, with a ROC, for example). Otherwise the R2 shows the proportion of variance explained by a variable of interest on another variable. Clarity on whether this research is about prediction versus explanation is fundamental and impacts the study design (see Shmueli 2010 famous paper on this). This point still does not come across.

We thank the reviewer for allowing us to clarify this issue. The aim of this work is to evaluate the bidirectional genetic relationship of perceived and objective social isolation with schizophrenia, using the following analyses: polygenic score, LDSC regression, genetic correlation, and Mendelian randomization models.

We use the term prediction in several instances throughout the manuscript when referring to polygenic score prediction. In a recent work, Plomin and Stumm support that the term prediction may be more appropriate than explanation in this context, as polygenic score prediction of behavioural traits are correlations and correlations do not imply causation (<https://psyarxiv.com/xbfmr/>).

We agree with the reviewer that there may be an implicit statistical incoherence of using the term “prediction” without reporting a predictive value. However, the use of polygenic scores as predictive tests has been discouraged considering that sensitivity and specificity of predictive models based on these variables is low. For instance, the area under the receiver operating curve for polygenic score prediction of schizophrenia risk in an independent SCZ-HC cohort in a recent schizophrenia GWAS study was only 0.62 (PMID:25056061). Therefore, reporting the predictive value of polygenic score prediction models using standard procedures such as ROC analyses may not be warranted. In our case, this is even more pronounced since cross-trait polygenic scores are used, which are expected to account for even less of the explained variance in an independent sample.

In order to clarify these aspects we have revised the manuscript to replace “polygenic score prediction” with “polygenic score contribution”. We have also clarified in the Methods section, that the term “prediction” is used in some instances throughout the manuscript to conform to the terminology used across the literature, but that these tests are not used with a predictive purpose, as follows:

“The term “prediction” is used in relation to polygenic score models to conform to standard terminology in the field. However, these models are not used with a predictive purpose.”

In this context, the associations presented are definitely significant, but the effect sizes small. this would need to be discussed (it suggests other variables, not considered here, may play a larger role both in prediction and in the causal/ explanatory pathways.

We thank the reviewer for this observation. Per the reviewer’s suggestion we have expanded our limitation paragraph in the Discussion section to mention this, as follows:

“Finally, the small effect sizes and low predictive values suggest that even if genetic variation may partially underpin the link between schizophrenia and LNL-ISO phenotypes, environmental variables are likely to play a substantial role in this association and should be explored in further epidemiological studies”

Please take the sentence on COVID 19 out, it is random, generic and not circumstantiated (it cites an opinion piece).

We thank the reviewer for this comment. Per the reviewer’s suggestion we have now removed this sentence from the paper.

Reviewer #3 (Remarks to the Author):

The authors have done a good job of addressing my concerns.

We thank the reviewer for these positive comments

I just have 2 more minor points.

Why is Nagelkerke's pseudo-R2 still reported in Fig 2A? This should be R2 on the liability scale, since the outcome variable is SCZ versus control status.

We thank the reviewer for allowing us to clarify this.

In our previous version we changed all pseudo-R2 to R2 based on liability scale per the reviewers' suggestion except in PGS predictions based on LNL-ISO phenotypes. We decided to keep Nagelkerke's pseudo-R2 for these analyses since there are no consistent prevalence estimates for the LNL-ISO constituent phenotypes (and even less so for LNL-ISO MTAG), which are required to estimate R2 based on the liability scale using the procedure described by Lee et al (2012).

We have now conducted a supplementary analysis that is shown in **Supplementary Figure 1** to estimate R2 based on the liability scale also for these phenotypes by using case-control definitions of LNL-ISO constituent phenotypes (described in **Supplementary Methods 2**) to estimate their prevalence in the UKBB population.

We have also modified the legend for Figure 2A to clarify this, as follows:

*"Figure 2A. PGS predictions of LNL-ISO and its constituent phenotypes (see legend) on an independent schizophrenia case-control sample (NSCZ = 1927; NHC = 1561). Explained variance attributable to PGS was calculated as the increase in Nagelkerke's pseudo-R2 between a linear model with and without the PGS variable. P values were obtained from the binomial logistic regression of the SCZ phenotype on PGS, accounting for Linkage Disequilibrium (LD) and including sex, age and 10 MDS ancestry components as covariates. Significant PGS predictions after FDR correction ($p_{FDR} < 0.05$) are marked with an asterisk. **See Supplementary Figure 1 for R2 values for PGS predictions on the liability scale estimated using UK Biobank prevalence for LNL-ISO constituent phenotypes.**"*

In Table 1, the column "Outliers" reports the number of pleiotropic variants removed from the MR estimate. I am not clear on which MR analysis this refers to.

We thank the reviewer for allowing us to clarify this. We have now clarified that the outliers correspond to the MR-PRESSO method by modifying the Table legend as follows:

"The column "Outliers" reports the number of pleiotropic variants removed with MR-PRESSO. MR-PRESSO β -Effects were estimated after removing the outliers."